# Defective DNA damage repair leads to frequent catastrophic genomic events in murine and human tumors

Manasi Ratnaparkhe[1], John K.L. Wong[2], Pei-Chi Wei[3], Mario Hlevnjak[2], Thorsten Kolb[2], Milena Simovic[1], Daniel Haag[4], Yashna Paul[2], Frauke Devens[2], Paul Northcott[5,6], David T.W. Jones[4], Marcel Kool[4], Anna Jauch[7], Agata Pastorczak[8], Wojciech Mlynarski[8], Andrey Korshunov[9], Rajiv Kumar [10], Susanna M. Downing[11], Stefan M. Pfister[4], Marc Zapatka[2], Peter J. McKinnon[11], Frederick W. Alt[3], Peter Lichter[2] & Aurélie Ernst[2]

Chromothripsis and chromoanasynthesis are catastrophic events leading to clustered genomic rearrangements. Whole-genome sequencing revealed frequent complex genomic rearrangements ($n = 16/26$) in brain tumors developing in mice deficient for factors involved in homologous-recombination-repair or non-homologous-end-joining. Catastrophic events were tightly linked to *Myc/Mycn* amplification, with increased DNA damage and inefficient apoptotic response already observable at early postnatal stages. Inhibition of repair processes and comparison of the mouse tumors with human medulloblastomas ($n = 68$) and glioblastomas ($n = 32$) identified chromothripsis as associated with *MYC/MYCN* gains and with DNA repair deficiencies, pointing towards therapeutic opportunities to target DNA repair defects in tumors with complex genomic rearrangements.

[1] Division of Molecular Genetics, German Cancer Consortium (DKTK), German Cancer Research Center (DKFZ); Faculty of Biosciences, Heidelberg University Germany, Heidelberg, 69120 Germany. [2] Division of Molecular Genetics, German Cancer Consortium (DKTK), German Cancer Research Center (DKFZ), Heidelberg, 69120 Germany. [3] Boston Children's Hospital, Howard Hughes Medical Institute and Department of Genetics, Harvard Medical School, Boston, 02115 MA, USA. [4] Hopp Children's Cancer Center at the NCT Heidelberg (KiTZ), Heidelberg, 69120 Germany. [5] Division of Pediatric Neurooncology, German Cancer Research Center (DKFZ), Heidelberg, 69120 Germany. [6] Department of Developmental Neurobiology, St. Jude Children's Research Hospital, Memphis, TN 38105-3678 United States. [7] Institute of Human Genetics, University of Heidelberg, Heidelberg, 69120 Germany. [8] Department of Pediatrics, Oncology, Hematology and Diabetology, Medical University of Lodz, Lodz, 91-738 Poland. [9] Clinical Cooperation Unit Neuropathology, German Cancer Research Center (DKFZ), Department of Neuropathology, Heidelberg University Hospital and German Cancer Consortium (DKTK), Heidelberg, 69120 Germany. [10] Division of Molecular Genetic Epidemiology; German Consortium for Translational Cancer Research (DKTK), German Cancer Research Center, Heidelberg, 69120 Germany. [11] Department of Genetics, St. Jude Children's Research Hospital, Memphis, 38105-3678 TN, USA. These authors contributed equally: Manasi Ratnaparkhe, John K.L. Wong. Correspondence and requests for materials should be addressed to A.E. (email: a.ernst@dkfz.de)

Chromothripsis and chromoanasynthesis are two forms of genomic instability that lead to complex genomic rearrangements affecting one or very few chromosomes[1–3]. These two types of catastrophic events play a role in numerous tumor entities, as well as in some congenital diseases[3,4]. The first form, chromothripsis, is characterized by the simultaneous occurrence of tens to hundreds of clustered DNA double-strand breaks[1,5]. The DNA fragments resulting from this shattering event are re-ligated by error-prone repair processes, with some of the fragments being lost. The outcome is a highly rearranged derivative chromosome, with oscillations between two or three copy number states[6]. Conversely, the local rearrangements arising from chromoanasynthesis exhibit altered copy number due to serial microhomology-mediated template switching during DNA replication[2]. Resynthesis of fragments from one chromatid and frequent insertions of short sequences between the rearrangement junctions are associated with copy number gains and retention of heterozygosity[2].

The availability of murine tumor models recapitulating these phenomena would substantially facilitate the investigation of the mechanistic aspects underlying complex genome rearrangements. We showed previously the role of constitutive and somatic DNA repair defects in catastrophic genomic events in the context of *TP53* and *ATM* mutations[5,7]. Further factors essential to the biochemical and signaling context of occurrence of these catastrophic events remain to be identified, and the role of repair processes in complex genome rearrangements needs to be better defined.

Homologous-recombination repair (HR) and canonical non-homologous end-joining (cNHEJ) represent the two major repair processes for DNA double-strand breaks in mammalian cells. Conditional inactivation of key repair factors of either of these two pathways, such as BRCA2 (HR), XRCC4, or Lig4 (cNHEJ) in nestin-expressing or *Emx1*-expressing neural progenitor cells leads to medulloblastomas (MBs) or high-grade gliomas (HGGs) in mouse models with a p53-deficient background[8,9] (Peter McKinnon, personal communication). We now show that these tumors frequently display complex genome rearrangements—also in the absence of cNHEJ—and that analyzing tumor development in these mice will help us to understand the role of repair processes in catastrophic genomic events.

## Results

**Inactivation of DNA repair factors essential for HR or cNHEJ leads to frequent complex genomic rearrangements.** Whole-genome sequencing of the tumors developing in BRCA2/p53, XRCC4/p53, or Lig4/p53-deficient animals showed frequent complex genomic rearrangements (Fig. 1a–c, Supplementary Figure 1), with a prevalence of 64% in the XRCC4/p53-deficient mice ($n = 11$ MBs), 60% in the Lig4/p53-deficient mice ($n = 5$ HGGs), 71% in the BRCA2/p53-deficient HGGs ($n = 7$), and 33% in the BRCA2/p53-deficient MBs ($n = 3$) (Fig. 1c, Supplementary Data 1). Our scoring criteria for complex genome rearrangements were based on the assessment of previously described hallmarks of catastrophic events[6] (e.g., clustering of breakpoints), using an algorithm identifying canonical chromothripsis with oscillations between two copy number states as well as events involving additional structural alterations[10], in combination with manual curation (see Methods section for details on the scoring criteria and Supplementary Data 2 for the output of the algorithm). This method also detects the many cases that do not display simple oscillations, such as partially oscillating copy number profiles with interspersed amplifications, and oscillations spanning multiple copy number levels due to aneuploidy[10]. Most cases showed rearrangement patterns dominated by segmental gains (see Fig. 1a, right panel), whereas two tumors in the BRCA2/p53-deficient animals were affected by complex rearrangements dominated by losses (Fig. 1a, left panel).

The high prevalence of catastrophic events in the tumors developing in these animals suggested that complex genomic rearrangements might represent driver events in tumorigenesis in these models. In line with this, murine orthologs of known MB and HGG oncogenes, such as *Myc, Mycn*, or *Cyclin D2* (in agreement with previous studies[8,9]) were frequently gained or amplified in association with the observed complex genomic rearrangements (Fig. 1d, Supplementary Figure 1, see Supplementary Data 1 for frequencies). These amplifications were linked with significantly increased expression levels of the respective oncogenes as compared to the expression values observed in the normal brain (Supplementary Figure 2a).

Importantly, the detected complex genomic rearrangements were not merely due to p53 deficiency, as re-analysis of published whole-genome sequencing data of a series of p53-deficient mouse models of MB[11,12] not based on DNA repair defects showed only 9/31 (29%) p53-deficient tumors with complex genomic rearrangements (Supplementary Figure 2b).

M-FISH analysis of MB cells from the XRCC4/p53-deficient mice showed that catastrophic events were associated with increased chromosome numbers (44–63 chromosomes per metaphase in tumors with complex genome rearrangements, as compared to 40–43 chromosomes per metaphase in tumors without catastrophic event, Fig. 2a). This finding is in line with the reported link between polyploidy and the detection of rearrangements due to one-off genomic catastrophic events[13,14]. In addition, tumor cells with complex genomic rearrangements showed higher total numbers of aberrations and frequently harbored marker chromosomes (Fig. 2b). Interestingly, chromosomes affected by complex genome rearrangements were also shown by a previous study to harbor recurrent breakpoints involved in translocations, deletions, and amplifications as demonstrated by CGH and FISH by Alt and colleagues in an independent set of tumors in XRCC4/p53-deficient mice[8].

From a global comparison of the genetic landscapes of the murine tumors developing in BRCA2/p53, XRCC4/p53, or Lig4/p53-deficient animals with human MBs and HGGs, respectively, the mouse tumors displayed a significantly lower mutational burden (Fig. 2c, right panel), as reported for mouse models in other tumor entities[15]. The frequency of structural variations was in the same range as for human HGGs with complex genome rearrangements and as for Sonic Hedgehog (SHH) MBs with *TP53* mutations (Fig. 2c).

**Amplifications of *Myc* or *Mycn* are linked with catastrophic events in XRCC4/p53 mice.** Strikingly, gains or focal amplifications of *Myc* or *Mycn* were detected in nearly all (10/11) MBs developing in the XRCC4/p53-deficient mice, in line with previous work[8] (Table 1, Supplementary Data 1). From these, 6/11 independently showed both gains of *Myc* or *Mycn* and regions of complex genome rearrangements, whereby the *Myc* or *Mycn* loci were occasionally directly included in the region affected by the catastrophic event (4/11, see Table 1 and Supplementary Data 1 and 2 for the exact regions involved). In one tumor, we detected complex genome rearrangements but no gain of *Myc* or *Mycn*; however, the *Mycn* expression level in this MB was comparable to the level observed in the tumors with gains of *Mycn* (Supplementary Figure 2). There was no tumor with neither complex genome rearrangement nor gain of *Myc* or *Mycn*, suggesting that these two events potentially play a driver role in tumor development in the context of cNHEJ and p53 deficiency in this entity.

In order to investigate further the putative link between MYC/MYCN and complex genomic rearrangements, we scored

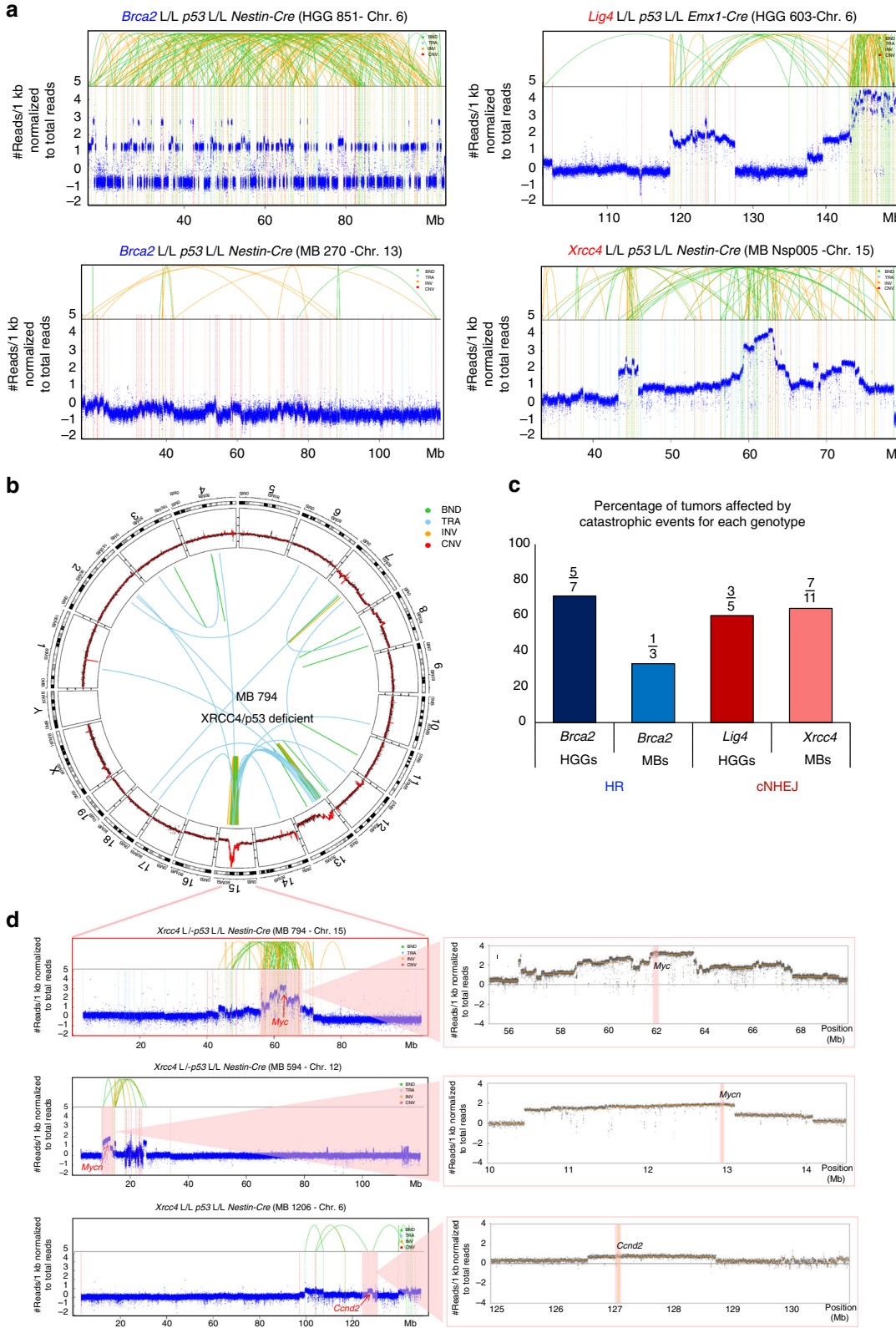

complex genome rearrangements in a series of *Myc*- or *Mycn*-based murine models of MB, respectively, for which whole-genome sequencing data were available from a previously published study[11]. In these MYC- or MYCN-driven models (with varying p53 status) only 1 out of 20 MBs showed a catastrophic event, suggesting that *Myc/Mycn* overexpression in neural progenitors in the absence of DNA repair defect does not lead to complex genomic rearrangements (Supplementary Figure 2b).

Depending on the context, *Myc/Mycn* amplifications may represent a consequence of the catastrophic event providing a strong selective advantage[5] or may possibly facilitate the

**Fig. 1** Inactivation of DNA repair factors essen for HR or cNHEJ in a p53 deficient background leads to frequent catastrophic events. **a** Examples of chromosomes affected by complex genome rearrangements for one tumor for each mouse genotype. HGG high-grade glioma; MB medulloblastoma. **b** Circos plot for a medulloblastoma showing complex genome rearrangements on chromosome 15 in a XRCC4/p53-deficient mouse. Magnified coverage plot for chromosome 15 is shown in **d** (upper panel). **c** Prevalence for catastrophic events in brain tumors developing due to inactivation of HR (blue, Brca2/ p53-deficient mice, n = 10) or cNHEJ (red, Lig4/p53 or XRCC4/p53-deficient mice, n = 16) in neural progenitors. **d** Complex genome rearrangements drive tumor development, as shown by the amplification of oncogenes associated with the catastrophic events. Blow-out views for all regions affected by complex genome rearrangements, circos plots and coverage plots for all chromosomes are presented in Supplementary Figure 1. BND, break end; TRA, translocation; INV, inversion; CNV, copy number variant

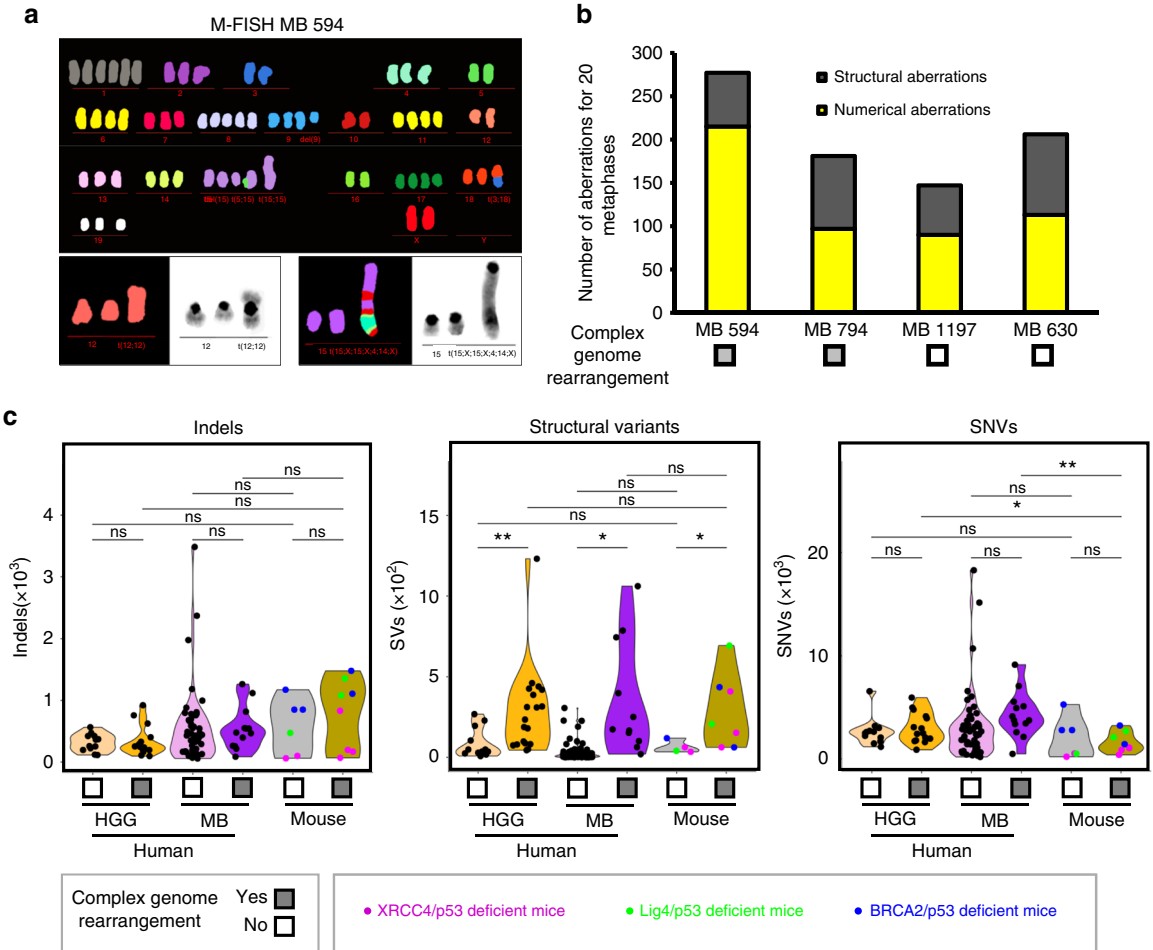

**Fig. 2** Karyotyping of the murine tumors and comparison of the global genetic landscapes with human tumors. **a** Complex genome rearrangements are associated with increased chromosome numbers and with structural aberrations. Left panel, M-FISH analysis of medulloblastoma cells from a XRCC4/p53-deficient mouse. **b** Quantification of the structural and numerical aberrations detected by M-FISH analyses for medulloblastoma cells derived from four Xrcc4/p53-deficient mice. **c** Indel, structural variant and SNV burdens by whole-genome sequencing in human high-grade gliomas (HGGs, n = 32) and medulloblastomas (SHH MBs, n = 68) as compared to murine HGGs and MBs (n = 14, colored according to the genotypes). Human SHH MBs with *TP53* mutations all show complex genome rearrangements (dark purple) *p < 0.05, **p < 0.01, beta-regression analysis

occurrence of complex genomic rearrangements, for instance due to increased levels of replication stress, in particular in combination with DNA repair deficiency.

These hypotheses prompted us to consider potential scenarios for the sequence of events between the gains of *Myc/Mycn* and complex genomic rearrangements by two-color FISH in two MBs (Supplementary Figure 3). In both tumors, two major clones were identified among the tumor cells, namely one clone with *Myc* or *Mycn* amplification, respectively, with co-occurrence of gains associated with complex genomic rearrangements, and a second clone with *Myc* or *Mycn* amplification but no gain associated with complex genomic rearrangements (Supplementary Figure 3). The sub-clonal events suggest two possibilities: (i)

the amplification of *Myc* or *Mycn* may occur before complex genomic rearrangements and possibly facilitate these events (ii) the *Myc/Mycn* gains (generated by the catastrophic event) may be retained and the complex genomic rearrangements lost (conceivably because the derivative chromosome did not offer any additional advantage).

Gains of *Myc* or *Mycn* were not as frequent in the HGGs from the BRCA2/p53 (2/7) or Lig4/p53 (2/5) deficient animals (Supplementary Data 1), making an analysis of the putative sequence of events challenging. However, the correlation between catastrophic events and gains of *Myc* or *Mycn* holds true in these tumors as well, with all HGGs showing a gain of either oncogene also displaying a catastrophic event.

**Table 1 Complex genome rearrangements and gains of *Myc/Mycn* in *Nestin-cre p53*[−/−] *Xrcc4*[−/−] mice)**

MBs (in *Nestin-cre p53*[−/−] *Xrcc4*[−/−] mice)

| MB ID | Complex genome rearrangement | *Mycn* gain (chr 12) | *Myc* gain (chr 15) |
|-------|------------------------------|----------------------|---------------------|
| 630 | No | Yes | Yes |
| 1187 | No | Yes | No |
| 1197 | No | Yes | No |
| 506 | No | Yes | No |
| 1206[a] | Chr. 6, 13 | No | No |
| 1207 | Chr. 7 | Yes | No |
| 1224 | Chr. 15 | Yes | Yes |
| 706 | Chr. 7 | Yes | No |
| 794 | Chr. 15 | Yes | Yes |
| 594 | Chr. 12 | Yes | No |
| NXP005 | Chr. 13, 15 | No | Yes |

[a]Expression levels for *Mycn* and *Myc* were comparable to MBs where gains of these loci were detected

**Increased DNA damage and inefficient apoptotic response may contribute to catastrophic events**. In order to understand what leads to catastrophic events in the precursor cells, we analyzed the cerebellum from early postnatal stages up to tumor onset in age-matched *Nestin-Cre p53*[−/−] *Xrcc4*[c/c], *Nestin-Cre p53*[−/−] *Xrcc4*[c/+] and *Nestin-Cre p53*[−/−] *Xrcc4*[c/−] animals (Fig. 3). No significant difference was detected between the genotypes regarding the proportions of Ki67 positive cells in the cerebellum or regarding telomere features, as evaluated by immunofluorescence staining and by telomere FISH combined with telomere content analysis of the sequencing data, respectively (Supplementary Figure 4). Loss of XRCC4 led to significantly higher levels of DNA DSBs at all examined developmental stages, as assessed by gH2AX and 53BP1 staining (Fig. 3, middle panels and Supplementary Figure 5). Therefore, alternative repair pathways seem to be unable to compensate entirely for the inactivation of cNHEJ in the neural tissue. Apoptosis was significantly less frequent in the cerebellum of *Nestin-Cre p53*[−/−] *Xrcc4*[c/−] as compared to *Nestin-Cre p53*[−/−] *Xrcc4*[c/+] animals at p8 and p80, as seen by TUNEL (Fig. 3, lower panel). The same trend was observed at p60 but the difference was not significant (Supplementary Figure 5). The increased DNA damage together with the less efficient apoptotic response in neural tissue likely contribute to the frequent occurrence of catastrophic events in MBs developing in XRCC4/p53-deficient mice.

Interestingly, loss of BRCA2 was reported to result in high numbers of DSBs in the cerebellum, as assessed by phosphorylated H2AX levels in *Brca2*[LoxP/LoxP];*Nestin-Cre* and *Brca2*[LoxP/LoxP];*Nes-cre;p53*[−/−] animals, as compared to *Brca2*[LoxP/LoxP] control mice[16]. Apoptosis was described as widespread throughout the *Brca2*[LoxP/LoxP];*Nestin-Cre* cerebellum, but attenuated when p53 is also inactivated[16]. Therefore, disruption of HR by deletion of *Brca2* in a p53-deficient background likely induces levels or types of DNA damage that cannot be repaired by other repair processes during neural development, similarly to what we observe in the context of cNHEJ deficiency.

**Repair processes involved in complex genomic rearrangements**. In order to investigate the DNA repair processes involved in catastrophic events, we analyzed the expression levels of essential factors in the main repair pathways in the MBs developing in XRCC4/p53-deficient mice, in the non-neoplastic cerebellum and in granule neural progenitors (GNPs) of control animals of the same genotype (Fig. 4a, left panel and Supplementary Figure 6). As XRCC4 is essential for efficient cNHEJ, this pathway is not functional in the *Xrcc4*-null neural progenitors, despite moderate

levels of Lig4. This highlights the occurrence of catastrophic events in the absence of cNHEJ and raises the question of which other pathways may play a role in the repair processes.

Key factors of HR were expressed at background levels in the normal cerebellum and in the GNPs but at moderate to high levels in the tumor cells (Fig. 4a, middle panel). Alternative end-joining factors were expressed at moderate levels in the normal cerebellum and in the GNPs and at significantly higher levels in the tumor cells (Fig. 4a, right panel), except Lig3, detected neither in the healthy cerebellum nor in the tumors in these animals. Consistent with this observation, Lig3 plays an essential role in mitochondrial DNA integrity but not in XRCC1-mediated nuclear DNA repair[17]. Altogether, as expected in the context of *Xrcc4* inactivation, HR at certain stages of the cell cycle (nearly absent in G1, most active in the S phase, and low in G2/M) but more likely alt-EJ are able to mediate the repair processes in the neural compartment of these animals. Interestingly, expression levels of repair factors in high-grade gliomas of BRCA2/p53 and Lig4/p53-deficient animals were very similar, with the exception of the two inactivated genes (Fig. 4b).

As an additional approach to the expression analysis, detailed comparisons of the microhomologies at the breakpoint junctions on the chromosomes affected by complex genomic rearrangements allow the inference of which repair processes were presumably active at the time of the catastrophic event (Fig. 4c, Supplementary Figure 6). In the tumors developing in the BRCA2/p53-deficient animals, we detected high proportions (close to 50%) of blunt ends and very short microhomologies (0–1 bp), consistent with a repair mediated by cNHEJ, which likely takes over in the absence of HR. For the XRCC4/p53 and Lig4/p53 animals, in contrast, the microhomologies were significantly longer (2–3 bp for 50–75% of the junctions), consistent with alternative end-joining and in agreement with previous studies[18]. This emphasizes the role of alternative end-joining in catastrophic events in the context of cNHEJ inactivation. Importantly, loss of p53 alone does not alter cNHEJ activity[18,19].

Mutational signatures can be used to evaluate deficiencies in specific repair processes. By adapting the method developed by Alexandrov and colleagues to assign specific signatures to mutational processes in human tumors[20], we identified mutational signatures contributing to somatic mutations in brain tumors developing in BRCA2/p53, Lig4/p53, and XRCC4/p53-deficient mice (Fig. 4d). Interestingly, all tumors developing in a context of cNHEJ/p53 inactivation displayed contributing proportions of signature 3 (associated with a failure of DNA DSB repair by HR[21]) in the same range as tumors from BRCA2/p53 animals and from a human MB occurring in a patient with biallelic inactivation of *BRCA2*. The contribution of mutational signature 3 in the context of functional HR may point to a novel etiology for signature 3. To further address this question, we performed mutational signature analysis for DNA-repair proficient murine tumors with wild-type p53 or with inactive p53/p53 related factors (e.g. CDKN2A). Even though further human and mouse tumors need to be analyzed to confirm these findings, our results point to a contribution of mutational signature 3 beyond HR deficiency (Supplementary Figure 6e). This signature may be linked with p53 deficiency, or possibly reflect consequences from excessive levels of DSBs and from the inability of the repair system to cope with high DNA damage, rather than necessarily pre-existing HR defects.

In order to show the dependence of the tumor cells on specific repair processes and to exploit DNA repair deficiencies, we treated the MB cells isolated from the *Nestin-Cre p53*[−/−]*Xrcc4*[−/−] mouse tumors with inhibitors of HR (B02 or Ri1, RAD51 inhibitors) and/or Alt-EJ (olaparib, PARP inhibitor) or induced DNA damage using topotecan (a topoisomerase 1 inhibitor), or applied a combination of all three. As controls, we applied the

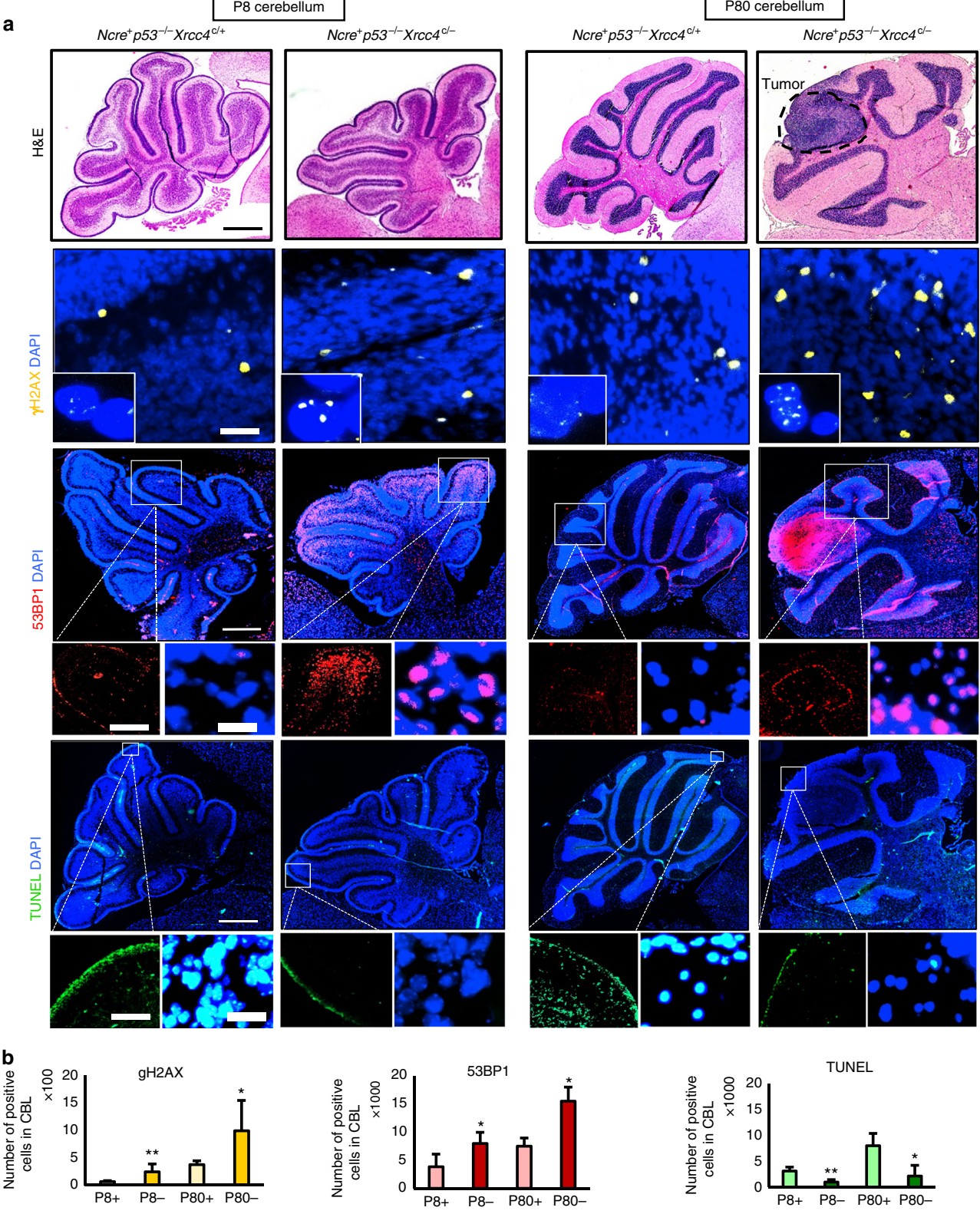

**Fig. 3** Analysis of brain tissue from early postnatal stages (P8) up to tumor formation (P80) in XRCC4/p53-deficient mice (*Nestin-cre p53$^{-/-}$ Xrcc4$^{c/-}$*) and control animals (*Nestin-cre p53$^{-/-}$ Xrcc4$^{c/+}$*). **a** Upper panels, H&E staining of cerebellum sections shows the location of the tumor at P80 (magnification, ×100). Middle and lower panels, representative immunofluorescence stainings for DSB markers gH2AX and 53BP1 and for terminal deoxynucleotidyl transferase (TdT)-mediated dUTP nick end labeling (magnification, ×100; insets, ×200, and ×600). Representative images for 11 analyzed cerebella are shown. Analysis of brain sections at intermediate stages (P60) are shown in Supplementary Figure 5. **b** Quantification of the number of cells positive for gH2AX, 53BP1, and TUNEL, respectively. CBL cerebellum; p8+, *p53$^{-/-}$ Xrcc4$^{c/+}$*, p8−, *p53$^{-/-}$ Xrcc4$^{c/-}$*. Between 4 and 12 sections per animal were analyzed, with 8 animals in the XRCC4-deficient group and 3 animals in the XRCC4-proficient group). Unpaired *t*-tests were done to test for significance

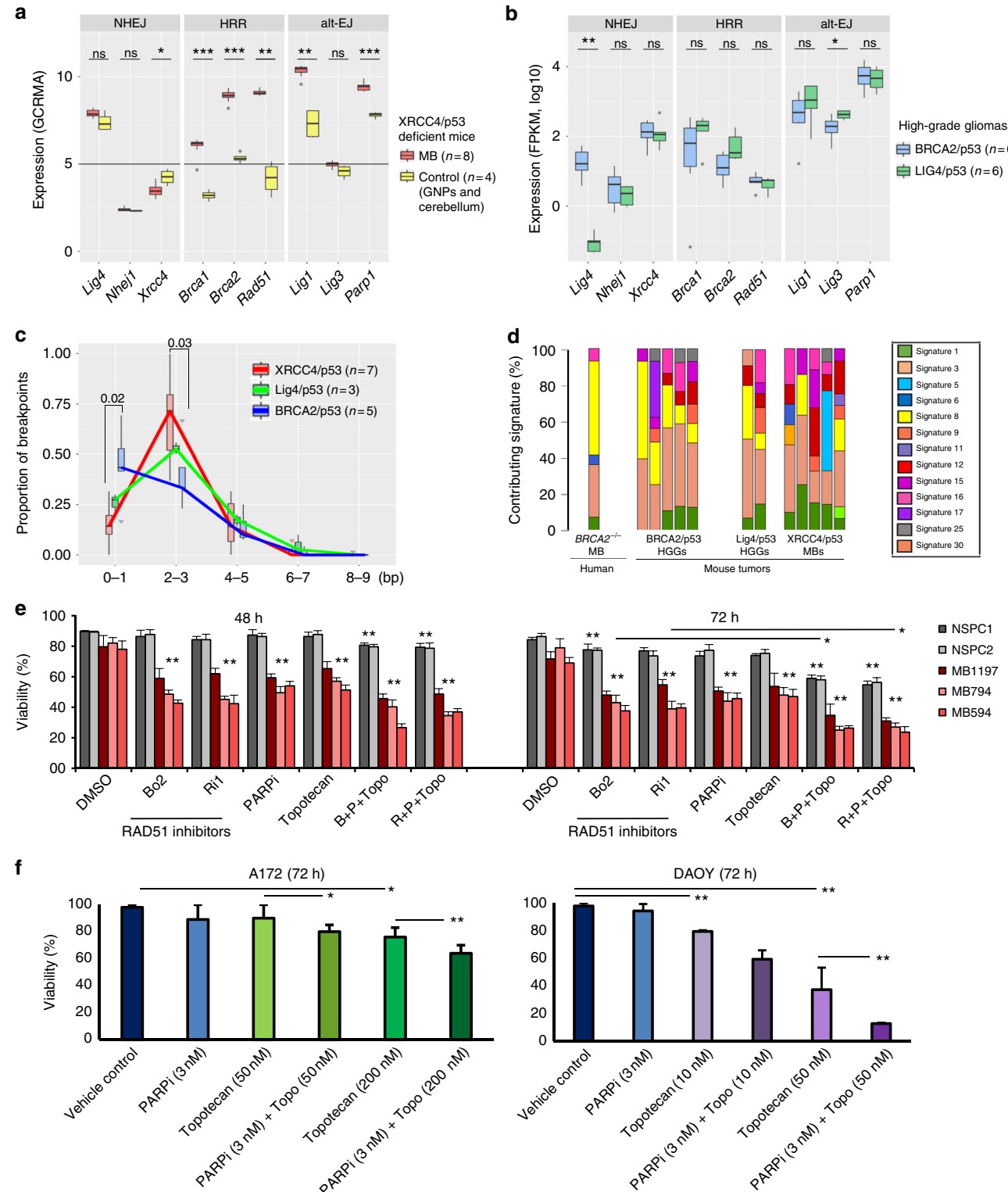

same treatments to neural progenitor cells from age-matched heterozygous animals (*Nestin-Cre p53*$^{-/-}$ *Xrcc4*$^{+/-}$). Combinations of RAD51 inhibitor, PARP inhibitor and topoisomerase inhibitor significantly decreased the viability of the MB cells without affecting the viability of the normal neural progenitor cells for short-term treatment (up to 48 h, Fig. 4e and Supplementary Figure 7). For longer treatments, the viability of the neural progenitor cells was reduced at the tested concentration for the combination treatment, but the fraction of viable cells

was still 30% higher than for the tumor cells. The combination treatments were significantly more efficient than the single treatments. Importantly, the doubling times of the control neural progenitor cells were not higher than those of the MB cells, showing that the observed difference in treatment response is not due to the respective proliferation rates.

To examine the potential therapeutic vulnerability of human MB and HGG cells with complex genome rearrangements, we applied the same strategy for two human cell lines. From 3 PARP

**Fig. 4** DNA repair pathways in brain tumors with catastrophic events. **a**, **b** Normalized expression levels for repair factors involved in cNHEJ, HRR and Alt-EJ. **a** Red, expression levels in medulloblastoma cells of XRCC4/p53 mice ($n = 8$), yellow, expression levels in the healthy cerebellum and granule neural progenitors ($n = 4$) for age and genotype-matched animals (expression levels for the non-neoplastic cerebellum and for granule neural progenitors were not significantly different and therefore shown as one box). The horizontal line depicts the threshold for which genes are considered to be expressed. **b** Normalized expression levels for repair factors in gliomas for BRCA2/p53 (blue, $n = 6$) or Lig4/p53 (green, $n = 6$) deficient animals. Unpaired $t$-tests were used to test for significance in **a** and **b**. **c** Analysis of the microhomology length at the breakpoints due to catastrophic events in tumors developing in XRCC4/p53, Lig4/p53 and BRCA2/p53 animals. The analysis focuses on chromosome regions affected by the catastrophic events. Beta-regression analysis was used to test for significance. **d** Mutational processes are dominated by signature 3 (associated with HR deficiency). The main contributing mutational signatures are very similar in tumors based on inactivation of factors essential for HR or for cNHEJ. A human MB from a *BRCA2* compound heterozygous patient is shown for comparison. **e** Synthetic lethality approaches targeting medulloblastoma cells from XRCC4/p53-deficient mice. Medulloblastoma cells (MB 594, MB 794, MB 1197) from *Nestin-Cre p53$^{−/−}$ Xrcc4$^{c/−}$* and *Nestin-Cre p53$^{−/−}$ Xrcc4$^{c/c}$* mice as well as control neural progenitor cells (NSPC1, NSPC2) from *Nestin-Cre p53$^{−/−}$ Xrcc4$^{+/−}$* mice were treated with RAD51 inhibitors (B02 or Ri1, 10 μM), PARP inhibitor (Olaparib, 5 μM), topotecan (100 nM) or with combinations. **f** Combination of PARP inhibitor with topoisomerase inhibitors reduces cell viability in human medulloblastoma (DAOY) and high-grade glioma cells (A172) with complex genome rearrangements. Cell viability is shown as a percentage of the viability for vehicle control cells in **e** and **f** (mean values and standard deviations for at least three independent experiments). ANOVA was used to test for significance in **e** and **f**. The medians for boxplots are indicated by center lines, rectangles span the interquartile ranges (whiskers show maximum and minimum)

inhibitors in clinical development for pediatric cancer, talazoparib was reported to show the highest efficacy[22]. Therefore, we used this PARP inhibitor in combination with topoisomerase inhibitors, which led to a significant reduction in cell viability for both lines (Fig. 4f and Supplementary Figure 7).

**Catastrophic events in human brain tumors**. To investigate the role of cNHEJ in catastrophic events in human neural cells, we used the CRISPR/Cas9 system to inactivate *TP53* and *LIG4* in iPSC-derived neural progenitors. We induced DNA DSBs by topoisomerase inhibitor treatment and analyzed the levels of DNA damage and apoptosis by gH2AX staining and TUNEL, respectively. After CRISPR/Cas9-mediated disruption of *TP53* and *LIG4*, we detected similar levels of DSBs in response to DNA damage but significantly less apoptotic cells as compared to wild-type control cells and to p53-deficient/*Lig4* wild-type cells (Fig. 5a, b and Supplementary Figure 8). Therefore, inactivation of DNA repair factors in combination with p53 deficiency might also contribute to catastrophic events in human neural precursor cells.

We next wanted to see how our findings on complex genome rearrangements in the mouse tumors relate to catastrophic events detected in human primary brain tumors. We first sequenced a medulloblastoma from a patient with Fanconi anemia due to compound heterozygosity in *BRCA2* (c.657_658delTG and c.7558C>T). Interestingly, this tumor showed complex genome rearrangements and focal amplification of *MYCN* (Fig. 5c). Beyond rare cases with germline mutations, we reanalyzed whole-genome sequencing data from human SHH medulloblastomas[23] ($n = 68$) and pediatric glioblastomas[24] ($n = 32$). Consistent with the link between MYC/MYCN and catastrophic events that we identified in the murine XRCC4/p53 MBs, gains or focal amplifications of the *MYC* or *MYCN* loci were detected at higher frequencies in human tumors with complex genome rearrangements as compared to cases without complex genome rearrangements, both in glioblastomas and in SHH MBs (Supplementary Data 2). In line with this result, a correlation between *MYC* amplification and chromothripsis was previously shown in group 3 MB[25]. In the human MBs and GBMs as in the mouse tumors, the chromosomes affected by the catastrophic events were not necessarily the chromosomes harboring the *MYC* or *MYCN* locus, respectively (Supplementary Data 1 and 2). Therefore, this association is not merely due to a selective advantage provided by the oncogene as a consequence of the massive rearrangement, but may in some instances be related to a possible facilitation of catastrophic events by *MYC* or *MYCN* activation. By a genome-wide search for amplified loci significantly associated with complex genome rearrangements across eight tumor entities, we

identified the *MYC* locus as strongly associated with catastrophic events (Supplementary Figure 9).

Importantly, analysis of the microhomologies at the breakpoints showed high proportions of the longer microhomologies (2–3 bp for close to 50% of the junctions) for breakpoints within the chromothriptic regions in human SHH MB (Fig. 5d), consistent with alternative end-joining and in agreement with our findings on the murine tumors. In human pediatric glioblastomas, the fraction of junctions with longer microhomologies was also higher as compared to tumors without chromothriptic event, but did not reach the significance level (Supplementary Figure 10a). Altogether, these findings point toward a role for alternative end-joining in catastrophic events in human brain tumors.

At the mutational signature level, a significant association between chromothripsis and mutational signatures 3 and 8, associated with DNA repair defects, was reported in SHH medulloblastoma[14]. We confirmed the link between DNA repair deficiency and complex genome rearrangements in additional tumor entities, namely adult glioblastoma ($n = 74$), breast cancer ($n = 356$), and melanoma ($n = 59$) (Supplementary Figures 10b-d). In glioblastoma, loss of *BRCA2*, *LIG4*, or *XRCC4* was associated with more frequent occurrence of complex genome rearrangements. In melanoma, we identified a significant association between mutational signature 3 (linked with HR deficiency) and complex genome rearrangements. In breast cancer, high homologous-recombination deficiency indexes were significantly associated with complex genome rearrangements. Therefore, inactivation of HR and NHEJ factors is linked with complex genome rearrangements across various tumor entities.

Constitutive inactivation of repair pathway components is infrequent in human brain tumors. We identified 49 MB or HGG patients with pathogenic germline mutations in DNA repair genes, a subset of these with biallelic inactivation (compound heterozygous or secondary somatic hit) from two previously published studies[14,26]. We observed significantly more complex genome rearrangements among the cases with biallelic inactivation of the repair factors, with 13/17 tumors showing catastrophic events having both alleles affected, as compared to 7/32 tumors without complex genome rearrangement displaying biallelic inactivation ($p = 0.0005$, Fisher exact test, Fig. 5e and Supplementary Data 4).

In analogy to the clinical use of PARP inhibitors in the context of BRCA-deficient breast cancer, our findings point toward therapeutic opportunities to target DNA repair defects in tumors with complex genomic rearrangements. Future studies will help defining the specific subsets of patients who may benefit from such a treatment.

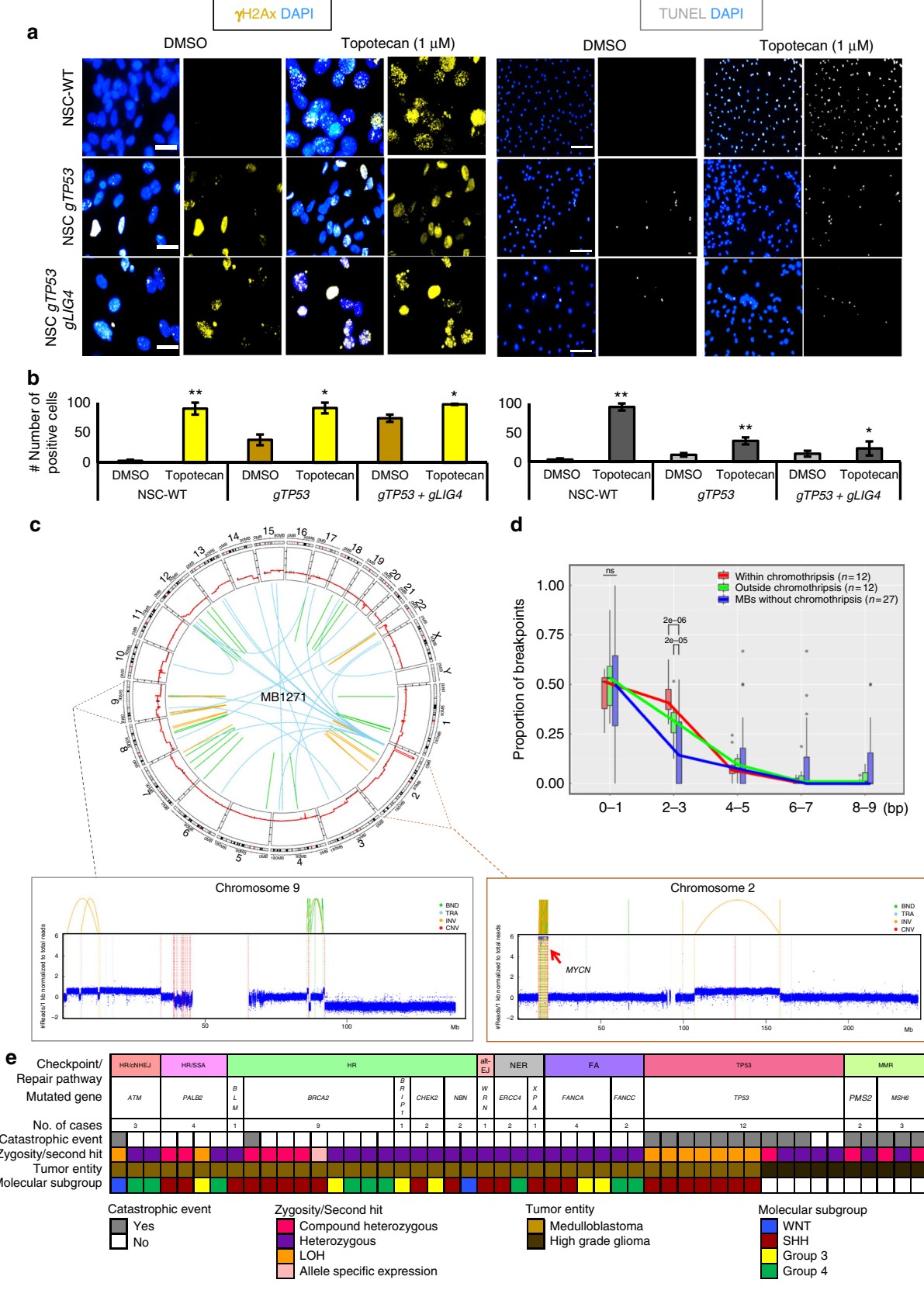

**Discussion**

We showed that inactivation of factors of DNA DSB repair by HR or NHEJ leads to frequent catastrophic genomic events in murine and human tumors. The availability of murine models recapitulating recently discovered forms of genome instability will facilitate the investigation of the mechanistic aspects underlying one-off complex genomic rearrangements. It will be important to distinguish between the contribution of p53 and the role of the repair factors themselves. As *Nestin-Cre Xrcc4* <sup>c/−</sup> p53<sup>+/+</sup>, *Nestin-Cre Xrcc4*<sup>c/−</sup> p53<sup>+/−</sup>, *Nestin-Cre Xrcc4*<sup>c/+</sup> p53<sup>+/−</sup>, *Nestin-Cre*

**Fig. 5** Analysis of human neural progenitor cells, medulloblastomas and high-grade gliomas with DNA repair defects. **a** Response to DNA damage after CRISPR/Cas9-mediated disruption of *TP53* and *LIG4* in human iPSC-derived neural progenitor cells. Left panel, immunofluorescence analysis of DSB marker gH2AX. Right panel, TUNEL. Topotecan treatment (1 μM) was applied for 48 h to induce DNA DSBs. **b** Quantifications of the numbers of positive cells in topotecan-treated and control cells after disruption of *TP53* and *LIG4*. Average values and standard deviations for three independent experiments are shown. Unpaired *t*-tests were used to test for statistical significance. **c** Chromothripsis in a human medulloblastoma with BRCA2 deficiency. Circos plot and coverage plots showing rearrangements associated with chromothripsis on chromosome 9 and *MYCN* amplification on chromosome 2. **d** Analysis of the length of the microhomologies in human SHH MBs at the breakpoints due to catastrophic events (red), on chromosome regions not affected by catastrophic events in the same tumors (green) or genome-wide in MBs not showing chromothripsis (blue). The median for each boxplot is indicated by the center line, the rectangle span the interquartile range (whiskers show maximum and minimum values). **e** Heatmap showing complex genome rearrangements in human medulloblastomas (*n* = 32) and high-grade gliomas (*n* = 17) with pathogenic germline mutations in major components of DNA repair and checkpoint pathways (re-analysis of whole-genome sequencing data from two published studies[14,26])

*Lig4 p53*[+/+], and *Nestin-Cre Lig4 p53*[+/−] animals do not develop medulloblastomas or gliomas[8,9] it is not feasible to address this question in the same genetic background. However, our scoring for complex genome rearrangements in a series of DNA-repair proficient p53-deficient mouse MBs suggests that deficiencies in repair factors in addition to the inactivation of p53 likely enhance the probability for catastrophic events. Further factors, as suggested by the high prevalence of catastrophic events in the mouse models based on *Rictor* inactivation for instance (Supplementary Figure 2b), remain to be identified.

Based on a cell culture system with inducible centromere inactivation followed by chromosome missegregation, fragmentation, and re-ligation[27], Ly et al. proposed that canonical NHEJ may be the predominant DNA repair process that mediates the re-joining of micronuclei-derived chromosome fragments. Inhibition of LIG4 or DNA-PKcs prevented reassembly of chromosomal fragments, whereas inhibition of HR or MMEJ by depletion of BRCA2/RAD51 or LIG3/PARP-1, respectively, had no effect on fragment reassembly[27]. In contrast, our study shows that cNHEJ is not the only process mediating the repair in complex genomic rearrangements, but that alt-EJ also plays an essential role. This discrepancy may be explained by the different functions and uses of repair processes in distinct cell types, to respond to various sources and extents of DNA damage.

In humans, chromothripsis was detected across a wide range of tumor entities, whereas chromoanasynthesis was mainly described in the context of congenital diseases[2,4]. No statistical framework has been developed to distinguish between chromothripsis and chromoanasynthesis, making an unambiguous distinction of both processes challenging. In the mouse tumors, we detected predominantly complex genome rearrangements dominated by segmental gains and rarely events dominated by losses (see Methods for details on the scoring procedures). Discrepancies in the prevalence of different types of catastrophic events between human and mouse tumors may relate to inter-species differences in terms of development of the neural lineages in the context of DNA repair deficiency, number of cell divisions, extent and type of exposure to external and internal triggers (e.g., replication stress), telomere biology, sensitivity to DNA damage or efficiency of specific repair processes and apoptotic response. Another potential explanation for this divergence may relate to differences in scoring methods and definitions for catastrophic events, and especially the fact that chromoanasynthesis was described later as compared to chromothripsis, implying that some of the initially reported chromothriptic events might now be referred to as chromoanasynthesis if assessed with hindsight.

Interestingly, we observed extreme examples of catastrophic events with exceptionally high numbers of breakpoints (Fig. 1a, top left panel) and other cases with lower numbers of breakpoints. Further work will be needed to understand whether the former are extreme outcomes of the same process, which are lethal for the cell in the vast majority of cases. Alternatively, the degree of the catastrophic event may also relate to the affected cell type, as the nestin compartment comprises a large population of different cells in the

developing brain. Depending on the specific cell affected by the event, the brain region, and the time point, the consequences may be tolerated or not, giving an advantage to a cell in a particular context. Our scoring for complex genome rearrangements in a series of previously characterized DNA-repair proficient murine MBs[11,12] identified catastrophic events only in SHH MBs and not in the other analyzed molecular subgroups of MB, supporting the cell-type-specific propensity for the acquisition of these catastrophic rearrangements.

We showed in the mouse models that, even though *Myc* overexpression alone is not sufficient to induce catastrophic events, amplifications of *Myc* and *Mycn* are linked with catastrophic events, and may not simply represent a consequence of these in all cases. Driving cell proliferation in the context of repair deficiency likely enhances the risk of chromosome fragmentation (e.g., as a consequence of chromosome segregation errors during mitosis and DNA damage on the missegregated chromosome) as well as DNA replication errors and collapsed replication forks, both processes which have been proposed to explain the rearrangements in chromothripsis and chromoanasynthesis[28]. The role of MYC and MYCN in replication stress[29] may represent a causative factor in the occurrence of catastrophic events. Interestingly, *MYCN* is a frequent driver gene in SHH MBs developed by Li-Fraumeni Syndrome patients, for which all analyzed tumors show chromothripsis[5]. Both for HGGs and for SHH MBs, the frequency of cases with complex genome rearrangements displaying gains of the *MYC* or *MYCN* loci was higher than in the tumors where no complex genome rearrangement was detected, pointing to a link between this oncogene and catastrophic events.

Surprisingly, we observed very similar mutational signatures between tumors developing in BRCA2/p53, Lig4/p53, and XRCC4/p53-deficient animals. In particular, mutational signature 3, described as associated with defects in HR, was detected in tumors from Lig4/p53 and XRCC4/p53 mice. Even though we cannot exclude a previously unrecognized role for XRCC4 and Lig4 in HR, the comparison with a number of murine tumors with wild type or mutant p53 suggests a possible link between signature 3 and p53 deficiency. Mutational signature 3 may point to mutational processes taking place when this major checkpoint is compromised, not necessarily due to HR defects. If it is validated in human tumors, this result may have important clinical implications, given the relevance of HR repair defects for therapeutic targeting, in particular for the use of PARP inhibitors based on BRCAness.

Altogether, our findings on the tight links between DNA repair deficiencies and catastrophic events and on synthetic lethality approaches bear the potential to identify targets for new therapeutic strategies for tumors with complex genomic rearrangements.

## Methods
**Library preparation and sequencing**. All tumors used for sequencing had a tumor cell content higher than 90% confirmed by neuropathological evaluation of the hematoxylin and eosin stainings. Human clinical samples and data were collected

after receiving written informed consent in accordance with the Declaration of Helsinki and approval from the respective institutional review boards. Purified DNA was quantified using the Qubit Broad Range double-stranded DNA assay (Life Technologies, Carlsbad, CA, USA). Genomic DNA was sheared using an S2 Ultrasonicator (Covaris, Woburn, MA, USA). Whole-genome sequencing and library preparations were performed according to the manufacturer's instructions (Illumina, San Diego, CA, USA or NEBNext, NEB). The quality of the libraries was assessed using a Bioanalyzer (Agilent, Stockport, UK). Sequencing was performed using the Illumina X Ten platform. Information on available sequencing data for all cases are summarized in Supplementary Data file 1.

**Cell culture**. NSPC1 and NSPC2 cell lines were grown as neurospheres and cultured in growth medium NBBG -Neurobasal A (Thermo Fisher, 10888–022), 2% B27- RA (Thermo Fisher, 12587–010), 0.5 mM glutamax (Thermo Fisher, 35050061), 5 µg/mL Gentamicin (VWR International, SAFSG1397) with 10 ng/mL EGF (Thermo Fisher, PHG0315), FGF (Thermo Fisher, PMG0035) and PDGF (Thermo Fisher, PMG0045). *Nestin-Cre p53−/− Xrcc4−/−* medulloblastoma cells derived from the murine tumors for MB 594, MB 630, MB 794, MB 1197, MB 1206, MB 1207, MB 1224, and NxP005 were all cultured and maintained as previously described[30]. All related animal work was performed under protocol 14–10–2790R approved by the Institutional Animal Care and Use Committee of Boston Children's Hospital. iPSC-derived neural progenitors were cultured and maintained as previously described[31]. DAOY and A172 cells were grown according to the ATCC recommendations.

**M-FISH**. M-FISH for cell lines MB 594, MB 630, MB 794, and MB 1197 was performed as described previously[32]. Briefly, seven pools of flow-sorted human chromosome painting probes were amplified and directly labeled using seven different fluorochromes (DEAC, FITC, Cy3, Cy3.5, Cy5, Cy5.5, and Cy7) using degenerative oligonucleotide primed PCR (DOP-PCR). Metaphase chromosomes immobilized on glass slides were denatured in 70% formamide/2xSSC pH 7.0 at 72 °C for 2 min followed by dehydration in a degraded ethanol series. Hybridization mixture containing labeled painting probes, an excess of unlabeled cot1 DNA, 50% formamide, 2xSSC, and 15% dextran sulfate were denatured for 7 min at 75 °C, pre-annealed at 37 °C for 20 min and hybridized at 37 °C to the denatured metaphase preparations. After 48 h the slides were washed in 2xSSC at room temperature for 3 × 5 min followed by two washes in 0.2xSSC/0.2% Tween-20 at 56 °C for 7 min, each. Metaphase spreads were counterstained with 4.6-diamidino-2-phenylindole (DAPI) and covered with antifade solution. Metaphase spreads were recorded using a DM RXA epifluorescence microscope (Leica Microsystems, Bensheim, Germany) equipped with a Sensys CCD camera (Photometrics, Tucson, AZ). Camera and microscope were used with the Leica Q-FISH software and images were processed on the basis of the Leica MCK software and presented as multicolor karyograms (Leica Microsystems Imaging solutions, Cambridge, United Kingdom).

**Nick translation and two-color FISH**. Nick translation was carried out for BAC clones all obtained from Source Bioscience of Myc (RP23 98D8), MycN (RP23 10C3), Grh1l (RP23 431C5), and Mtbp1 (RP23 288J22). Two-color FISH[33] was performed on metaphase spreads of MB 594 and MB 794 using fluorescein isothiocyanate-labeled probes and rhodamine-labeled probes. Pre-treatment of slides, hybridization, post-hybridization processing and signal detection were performed as described previously[33]. Samples showing sufficient FISH efficiency (>90% nuclei with signals) were evaluated. Signals were scored in, at least, 100 non-overlapping metaphases. Metaphase FISH for verifying clone-mapping position was performed using peripheral blood cell cultures of healthy donors as outlined previously.

**H&E staining, immunofluorescence, and TUNEL on tissue sections**. Hematoxylin and eosin (H&E) staining and immunofluorescence staining's were performed on 4 µm formalin-fixed paraffin-embedded mouse brain sections. H&E staining was evaluated by a neuropathologist. Tissue sections from each cohort of p8, p60, and p80 for at least three animals per genotype were stained. Sections were deparaffinized, antigen retrieval was performed in 10 mM citrate buffer pH 6.0 for 40 min and sections were cooled down to room temperature. Slides were then washed with PBS for 5 min and then blocked with blocking solution (10% goat serum diluted in PBS, 0.2% Triton X-100) for 1 h. Slides were then incubated with primary antibodies, gh2Ax (Cell Signaling, 2577S) 53BP1 (Santa Cruz, sc-22760), Ki67 (Abcam, 15580), all at 1:100 dilution overnight at 4 °C. The next day slides were washed three times with PBS for 10 min each and incubated for an hour at room temperature with rabbit secondary (Invitrogen, Alexa 488, A11034 or Life Technology, Alexa 568, A11036) at 1:1000 dilution. After the incubation, slides were washed with PBS three times for 10 min each and then were mounted with DAPI Fluromount (Southern Biotech, 0100–020) and image analysis of the mouse cerebellum was done with Axio Zeiss Imager.M2 microscope. For TUNEL, the manufacturer's protocol for paraffin-embedded slides was followed (Roche, 11684795910).

**Quantification of the number of positive cells**. Quantifications of the number of positive cells based on immunofluorescence analysis of mouse cerebellum tissue pictures were done using ImageJ software. In total, cerebellum tissue from 14 mice

was analyzed, with a minimum of three tissue sections for each brain and for each antibody. For each animal, cerebellum tissue sections from the left, middle and right side of the cerebellum were used for quantification purposes. The images were processed using an ImageJ macro designed by DKFZ Imaging core facility. In brief, after converting the picture into a 16 bit black and white picture, the macro first identifies the boundaries of the cerebellum. Background fluorescence is then subtracted, followed by Gaussian blur of sigma = 2. Minimum and maximum threshold for each of the stainings was selected and positive particles (cells) were analyzed by the macros resulting in the scoring of the numbers of positive cells (foci and pan-nuclear staining) in the cerebellum.

**Telomere FISH**. Telomere FISH (Agilent, K5325) was performed according to the manufacturer's protocol as well. Slides incubated with the probe were washed three times 10 min each with PBS and then were incubated with secondary antibody Alexa 568 (Thermo Fisher, A11036) at 1:1000 dilution for 1 h at room temperature. Slides were then washed again with PBS for 10 min each three times and were mounted with DAPI Fluromount (Southern Biotech, 0100–020). Image analysis was done with Axio Zeiss Imager.M2 microscope.

**Synthetic lethality experiments with murine cells**. Tumor cell lines MB 594, MB 794, and MB 1197, and control cell lines NSPC1 and NSCP2 were plated in 6-well plate for 10 h and then treated either with 10 µM B02 (Sigma-Aldrich, SML0364) or 10 µM Ri1 (Merck Millipore, 553514) or 5 µM Olaparib (Biozol, AZD2281) or 100 nM Topotecan (Apex Bio, B4982) or a combination of B02, Olaparib and Topotecan or a combination of Ri1, Olaparib and Topotecan at the above-mentioned concentrations for 24 h, 48 h, or 72 h. For each time point, cell viability was measured by cell viability analyzer Vi-Cell XR (Beckman Couter).

**Cytotoxicity assays on human cells**. Cells were plated at a density of 5000 cell per in 96-well plates (Corning, 3596) and treated with the media containing desired drug concentration (0–1000 nM, dissolved in DMSO) after 24 h. Cell viability was assayed after 72 h exposure to Talazoparib (Abmole, M1732), Topotecan (ApexBio, B4982), and Camptothecin (Biozol, S1288) alone or in combination. A quantity of 20 µl of 5 mg/ml MTT (Thiazolyl Blue Tetrazolium Bromide, Sigma-Aldrich, M5655) was added to the media and incubated for 4 h at 37 °C. The medium was completely removed and formazan crystals dissolved in DMSO. Absorbance was measured at 560 nm using microplate reader (Mithras LB 940, Berthold technologies). Values from the blank measurements were subtracted from the average cell viability based on four technical replicates and normalized to vehicle control using GraphPad Prism software (GraphPad Software Inc.). A minimum of three independent experiments were done.

**CRISPR-cas targeted gene disruption**. Guide RNAs for *TP53* and *LIG4* were constructed and cloned into lenti CRISPR v-2 (Addgene, 52961) according to the original online protocol of the Zhang lab (http://www.genome-engineering.org/crispr/wp-content/uploads/2014/05/CRISPR-Reagent-Description-Rev20140509.pdf). Following genes were targeted:
   *TP53* (gRNA2:CGACCAGCAGCTCCTACACCGG)
   *LIG4* (gRNA2:CGTCTGAGTTATAAGTTGAAGA)
   and (gRNA3:CGGCTTATACGGATGATCATAA). Virus production and transduction were done as described previously[34]; in brief, pLentiV2, pDMDG.2, and pSPAX were co-transfected in HEK293T cells, and virus-containing supernatant was concentrated by ultra-centrifugation. Transduction was done by adding concentrated virus particles to both NSC lines for 24 h, after which cells were maintained under selection either with puromycin or blasticidin at 2 µg/mL for 2–4 weeks. For CRISPR-mediated disruption of *TP53*, an additional selection for functional knockout was done using 20 µM nutlin treatment. After selection, cell lysates were made for western blotting or cells were grown for further experiments.

**Western blotting**. For western blot experiments, NSC cells were collected utilizing cell scrapers. Supernatant and scraped cells were pooled and washed three times with ice-cold PBS containing Roche Complete protease and PhosSTOP phosphatase inhibitors (Roche, 04693132001, 04906845001). For all lines, the washed pellet was briefly re-suspended in laemmli buffer containing benzonase (Novagen, 70664–3) and boiled at 95 °C for 5 min. Protein concentration was estimated using BCA. A total amount of 30 µg protein was loaded per lane on pre-casted SDS gels (Novex, NP0303BOX) and separated according to the manufacturer's instructions. Immunoblotting was done either on PVDF membranes in a tank blot system, using a borate-based buffer system (25 mM sodium borate, 1 mM EDTA, pH 8.8), or on nitrocellulose utilizing a semidry blotting machine with CAPSO buffer (10 mM CAPSO adjusted to pH 11). Membranes were blocked with 5% milk powder in TBST for 1 h and probed with the primary antibodies like Alpha-Tubulin (Abcam, 52866) 1:2000, LIG4 (sc271299) 1:500, TP53 (Progen, 65139) 1:5000. Incubation was performed overnight under constant agitation at 4 °C. Membranes were washed with TBST and incubated for 30 min with HRP coupled secondary anti-mouse or anti-rabbit antibodies (Jackson Immuno Research Laboratories, 115-035- 003 and 111-035-144) 1:1000. After washing, detection was done using enhanced chemiluminescence and images were recorded with an Odyssey Fc or Bio-rad Imaging System (LI-COR Biotechnology). Quantification was done with

the built-in quantification software. All signals were normalized to the respective tubulin signal of the lane.

**Immunofluorescence for cultured cells**. Immunofluorescence was done as previously described[7]. In brief, cells were grown on coverslips in 6-well plates and treated with Topotecan (1 μM) or DMSO (control) for 48 h, fixed and stained with cold methanol for 5 min and then in cold acetone for 30 s. Slides were incubated with 10% goat serum for 1 h followed by primary antibody staining for gH2AX, 53BP1 at 1:400 dilution for 1 h. Slides were washed three times with PBS for 10 min each and incubated for 1 h at room temperature with rabbit secondary antibody (Alexa 488 or Alexa 568) at 1:1000 dilution. After the incubation, slides were washed with PBS three times for 10 min each and then mounted with DAPI Fluromountor slides were stained for TUNEL (according to manufacturer's protocol). Image analysis was done with Axio Zeiss Imager.M2 microscope. For each of the three independent experiments, 100 cells were counted per staining to score for positive cells.

**Validation of a subset of breakpoints by PCR**. Genomic regions surrounding predicted breakpoints were amplified by conventional PCR. Primers for 14 breakpoints for MBNxp005 and MB851 were designed and PCR was done as described[35]. In brief PCR was carried out with JumpStart™ REDAccuTaq® LA DNA Polymerase (Sigma-Aldrich Inc., St. Louis, MO) in a 50 μl reaction volume and with 20 ng of genomic DNA as template. The following program was used: initial denaturation at 94 °C for 30 s, followed by 25 cycles of a 3-Step-Touchdown: 1. (94 °C 5 s, 68 °C 30 s, 68 °C 6 min), 2. (94 °C 5 s, 66 °C 30 s, 68 °C 6 min), 3. (94 °C 5 s, 64 °C 30 s, 68 °C 6 min); followed by an additional cycle of 68 °C 30 min. Fragments were visualized by gel electrophoresis.

**Statistics**. Statistical analysis was performed for all the experiments and data are presented as means ± SD unless otherwise stated. Unless otherwise specifically stated in the legends, all comparisons between different groups for immunofluorescence analysis was made using unpaired Student's tests. For drug response experiments, ANOVA was used. For SV/Homology/Indel size analysis, Beta regression was used. Respective statistical analysis for each figure can be found in the legends. $P$-values of 0.05 or lower were considered to be statistically significant for all experiments.

**Bioinformatic analysis**. Whole-genome sequencing, quality check and alignment workflow: We performed paired end 150 bp high coverage (~×50) whole-genome sequencing on the HiSeq X-Ten. Sequencing reads were checked by FastQC for base quality, duplication levels, GC bias and excess primer sequences. All samples passed the sequence quality checks. Afterwards, sequencing reads were aligned to the mouse genome assembly mm10 by the Burrows-Wheeler Aligner (BWA-MEM v0.7.8). Sorting and mark duplication of aligned reads were performed by Sambamba v0.5.9. Subsequent alignment files were assessed by samtools (v1.7) flagstats for mappability information and the percentage of properly paired reads. Samples with less than 70% properly paired reads and a bimodal insert size distribution were removed from the analysis.

Single nucleotide variant (SNV) and InDel calling: For murine tumors, SNVs and InDels were called by GATK (v.4.0.5.1) Mutect2[36] with mouse dbSNP build 142. Mutect2 involves two Bayesian classifiers to identify high confidence candidate somatic mutations at a given genomic locus using tumor and matched normal BAM files. A panel of normal controls was built using 14 high quality mouse tails as a blacklist for somatic SNVs/InDels. Unpaired mice (for which no matched germline controls were available) were excluded from the analysis. The resulting SNVs with quality filter PASS were used for downstream analysis.

For human tumors, SNVs and InDels were called by two different tools, by both GATK (v.4.0.5.1) Mutect2 and the DKFZ SNVs/InDels workflow described previously[14]. SNVs and InDels from Mutect2 were only used for the mutational loading comparison between mouse and human. Output from DKFZ SNVs workflow were used for mutational signatures analysis.

Detection of somatic structural variants (SVs): To maximize the sensitivity and specificity for microhomology detection at the structural variant junctions, assembly based detection methods were preferred over discordant read pair methods. Two assembly based SV detection methods, Svaba[37] and novoBreak[38] were used to detect the accurate breakpoints of the SVs. The controls used for subtraction of germline SVs were selected depending on the availability of paired specimens, either (A) matched germline control or (B) pooled controls were used. The pooled control strategy is only used for highly inbred mice samples. Additional hard filtering of novoBreak outputs was applied to reduce false positives by the following criteria: mapping quality (>= 30) and minimum coverage (>= 2). Svaba outputs were selected for most tumors unless the pooled controls strategy failed. The quality is considered to be poor when there are large amounts (>50%) of small SVs (smaller than 500 bp) detected by Svaba but not in the novoBreak output. For one mouse tumor, the novoBreak output had to be used. Intrachromosomal rearrangements in the same orientation were annotated as break end (BND) as their connection with copy number changes is difficult to resolve in scenarios of complex rearrangements.

Sensitivity of detection for complex genome rearrangements: To detect a shift in copy number of 0.1 (on a log2 scale), it is possible if it is a gain or loss present even in only 10% of the cells. This calculation is based on a tumor purity close to 100%, a diploid genome, a coverage of ×30, a low noise and with bins of 4 kb.

Microhomology analysis for structural variants (SVs): Microhomology sizes were extracted from precisely mapped SVs detected by the assembly method of Svaba. Samples without available Svaba output were excluded from the microhomology analysis. To investigate whether particular microhomology lengths were preferred for DSB repair in each tumor sample, the frequencies of each microhomology bins were compared. The analysis was focused on junctions with blunt ends or with microhomologies within 1–9 bp length. Junctions fulfilling these criteria were extracted and grouped into 5 different categories (0–1 bp, 2–3 bp, etc.). Samples with less than 5 breakpoints within the interested region were excluded from the analysis. To assess if breakpoints could be repaired by a different pathway within the region affected by complex genome rearrangements, only structural variants within the predefined chromothriptic region were included in the test.

Copy number variant and loss-of-heterozygosity detection: In the human cohort, ACESeq was used for the copy number profiling and the loss-of-heterozygosity detection (see Kleinheinz, K. et al. ACESeq - allele specific copy number estimation from whole-genome sequencing. https://github.com/DKFZ-ODCF/ACEseqWorkflow). We have scanned each gene candidate according to UCSC hg19 gene definition and evaluated if the gene loci are affected by any copy number variants.

For the mouse cohort, CNVkit (v.0.9.3) was used for copy number segmentation and profiling. All available germline mouse controls were pooled as the normal reference for copy number comparison. Genome wide coverage for each mouse tumor were sampled per coverage bin of ~1000 bp, and subsequently compared with the normal reference to retrieve the copy number status. Due to the high level of inbreeding of the mice cohort, it was not possible to perform BAF/loss-of-heterozygosity analysis.

Criteria for scoring for complex genome rearrangements: The SVs and CNVs detected by the above-mentioned methods were used for in silico scoring for complex genome rearrangements by Shatterseek[10], which allows statistical testing of the different criteria. Chromosomes marked in yellow in Supplementary Data 1 (output of the algorithm) show complex genome rearrangements based on pval_exp_cluster. Manual curation by visual inspection of zoom-in plots (Supplementary Figure 1) using combined CNVs and SVs information was done and excluded events with less than 10 copy number switches within 50 Mb.

Expression array analysis: The R library 'affy' was used to process the raw CEL files of the Affymetrix Mouse 430 v2.0 expression array. For better expression level comparison between genes, GCRMA was used for the normalization of probe-set expression values. GCRMA considers the sequence information of probes to adjust for the binding affinity bias of each probe. To retrieve gene-based expression values, the average expression values of all highly specific probes for a gene were taken as the gene expression value.

Mutational signature analysis: Supervised mutational signature analysis of high-confidence somatic SNVs in individual samples was performed using quadratic programming formalism as described previously[14]. The expanded set of 30 canonical mutational signatures was used for decomposition of somatic mutations (http://cancer.sanger.ac.uk/cosmic/signatures). Furthermore, canonical mutational signatures were re-normalized using the observed trinucleotide frequency in the mouse genome to the one of the human genome for mouse samples, and for human exome samples normalization was performed using the observed trinucleotide frequencies in the human exome as defined by the target region depending on the enrichment kit used. The fit of the deciphered mutational signatures were evaluated by the cosine similarity between the reconstructed 96-triplets composition and the original input. Samples were removed from the analysis if the similarity was below 0.8.

Telomere length analysis: The total telomere length for each mouse sample was estimated by Telomerehunter. Mapped and unmapped telomeric reads were extracted by 4 canonical telomeric repeats: TTAGGG, TCAGGG, TGAGGG, and TTGGGG. (see Feuerbach, L. et al. TelomereHunter: telomere content estimation and characterization from whole-genome sequencing data. www.dkfz.de/en/applied-bioinformatics/telomerehunter/telomerehunter.html, doi:10.1101/065532)

Genome visualization: Zoom in plots for specific regions were produced by the Sushi package in bioconductor[39]. Circos plots were produced by the bioconductor package 'circlize'.

## Data availability

The datasets generated and/or analyzed during the current study are available from the corresponding author on reasonable request. For the sequencing data and the array data, accession codes are SRA: PRJNA491653, GEO: GSE120344

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

# ARTICLE

3. Zhang, C. Z., Leibowitz, M. L. & Pellman, D. Chromothripsis and beyond: rapid genome evolution from complex chromosomal rearrangements. *Genes Dev.* **27**, 2513–2530 (2013).

4. Rode, A., Maass, K. K., Willmund, K. V., Lichter, P. & Ernst, A. Chromothripsis in cancer cells: an update. *Int. J. Cancer* **138**, 2322–2333 (2016).

5. Rausch, T. et al. Genome sequencing of pediatric medulloblastoma links catastrophic DNA rearrangements with TP53 mutations. *Cell* **148**, 59–71 (2012).

6. Korbel, J. O. & Campbell, P. J. Criteria for inference of chromothripsis in cancer genomes. *Cell* **152**, 1226–1236 (2013).

7. Ratnaparkhe, M. et al. Genomic profiling of acute lymphoblastic leukemia in ataxia telangiectasia patients reveals tight link between ATM mutations and chromothripsis. *Leukemia* **31**, 2048-2056 (2017).

8. Yan, C. T. et al. XRCC4 suppresses medulloblastomas with recurrent translocations in p53-deficient mice. *Proc. Natl Acad. Sci. USA* **103**, 7378–7383 (2006).

9. Frappart, P. O. et al. Recurrent genomic alterations characterize medulloblastoma arising from DNA double-strand break repair deficiency. *Proc. Natl Acad. Sci. USA* **106**, 1880–1885 (2009).

10. Cancer Genome Atlas, N. Genomic classification of cutaneous melanoma. *Cell* **161**, 1681–1696 (2015).

11. Poschl, J. et al. Genomic and transcriptomic analyses match medulloblastoma mouse models to their human counterparts. *Acta Neuropathol.* **128**, 123–136 (2014).

12. Akgul, S. et al. Opposing tumor-promoting and -suppressive functions of rictor/mTORC2 signaling in adult glioma and pediatric SHH medulloblastoma. *Cell Rep.* **24**, 463–478 e465 (2018).

13. Mardin, B. R. et al. A cell-based model system links chromothripsis with hyperploidy. *Mol. Syst. Biol.* **11**, 828 (2015).

14. Grobner, S. N. et al. The landscape of genomic alterations across childhood cancers. *Nature* **555**, 321-327 (2018).

15. Mueller, S. et al. Evolutionary routes and KRAS dosage define pancreatic cancer phenotypes. *Nature* **554**, 62–68 (2018).

16. Frappart, P. O., Lee, Y., Lamont, J. & McKinnon, P. J. BRCA2 is required for neurogenesis and suppression of medulloblastoma. *EMBO J.* **26**, 2732–2742 (2007).

17. Simsek, D. et al. Crucial role for DNA ligase III in mitochondria but not in Xrcc1-dependent repair. *Nature* **471**, 245–248 (2011).

18. Wei, P. C. et al. Long neural genes harbor recurrent DNA break clusters in neural stem/progenitor cells. *Cell* **164**, 644–655 (2016).

19. Panchakshari, R. A. et al. DNA double-strand break response factors influence end-joining features of IgH class switch and general translocation junctions. *Proc. Natl Acad. Sci. USA* **115**, 762-767 (2018).

20. Alexandrov, L. B. et al. Signatures of mutational processes in human cancer. *Nature* **500**, 415–421 (2013).

21. Nik-Zainal, S. et al. Landscape of somatic mutations in 560 breast cancer whole-genome sequences. *Nature* **534**, 47–54 (2016).

22. Stewart, E. et al. Targeting the DNA repair pathway in Ewing sarcoma. *Cell Rep.* **9**, 829–841 (2014).

23. Jones, D. T. et al. Dissecting the genomic complexity underlying medulloblastoma. *Nature* **488**, 100–105 (2012).

24. International Cancer Genome Consortium PedBrain Tumor, P. Recurrent MET fusion genes represent a drug target in pediatric glioblastoma. *Nat. Med.* **22**, 1314–1320 (2016).

25. Northcott, P. A. et al. Subgroup-specific structural variation across 1,000 medulloblastoma genomes. *Nature* **488**, 49–56 (2012).

26. Waszak, S. M. et al. Spectrum and prevalence of genetic predisposition in medulloblastoma: a retrospective genetic study and prospective validation in a clinical trial cohort. *Lancet Oncol.* **19**, 785–798 (2018).

27. Ly, P. et al. Selective Y centromere inactivation triggers chromosome shattering in micronuclei and repair by non-homologous end joining. *Nat. Cell Biol.* **19**, 68–75 (2017).

28. Zhang, C. Z. et al. Chromothripsis from DNA damage in micronuclei. *Nature* **522**, 179–184 (2015).

29. Maya-Mendoza, A. et al. Myc and Ras oncogenes engage different energy metabolism programs and evoke distinct patterns of oxidative and DNA replication stress. *Mol. Oncol.* **9**, 601–616 (2015).

30. Huang, X., Ketova, T., Litingtung, Y. & Chiang, C. Isolation, enrichment, and maintenance of medulloblastoma stem cells. *J. Vis. Exp.* https://doi.org/10.3791/2086 (2010).

31. Palm, T. et al. Rapid and robust generation of long-term self-renewing human neural stem cells with the ability to generate mature astroglia. *Sci. Rep.* **5**, 16321 (2015).

32. Geigl, J. B., Uhrig, S. & Speicher, M. R. Multiplex-fluorescence in situ hybridization for chromosome karyotyping. *Nat. Protoc.* **1**, 1172–1184 (2006).

33. Lichter, P. et al. High-resolution mapping of human chromosome 11 by in situ hybridization with cosmid clones. *Science* **247**, 64–69 (1990).

34. Phillips, E. et al. Targeting atypical protein kinase C iota reduces viability in glioblastoma stem-like cells via a notch signaling mechanism. *Int. J. Cancer* **139**, 1776–1787 (2016).

35. Korbel, J. O. et al. Paired-end mapping reveals extensive structural variation in the human genome. *Science* **318**, 420–426 (2007).

36. Cibulskis, K. et al. Sensitive detection of somatic point mutations in impure and heterogeneous cancer samples. *Nat. Biotechnol.* **31**, 213–219 (2013).

37. Wala, J. A. et al. SvABA: genome-wide detection of structural variants and indels by local assembly. *Genome Res.* **28**, 581–591 (2018).

38. Chong, Z. et al. novoBreak: local assembly for breakpoint detection in cancer genomes. *Nat. Methods* **14**, 65–67 (2017).

39. Phanstiel, D. H., Boyle, A. P., Araya, C. L. & Snyder, M. P. Sushi.R: flexible, quantitative and integrative genomic visualizations for publication-quality multi-panel figures. *Bioinformatics* **30**, 2808–2810 (2014).

## Acknowledgements

We would like to thank Michaela Hergt and the tissue bank of the National Center for Tumor Diseases (NCT, Heidelberg, Germany) for technical assistance, Achim Stephan for help with library preparation, Laura Siebert and Norman Mack for advice on paraffin embedding, Andrea Wittmann for help with the preparation of metaphase spreads, and Brigitte Schoell for M-FISH analyses. We would also like to express our gratitude to Angela Schulz, Nicolle Diessl, Laura-Jane Behl, Stephan Wolf from the DKFZ Genomics and Proteomics Core Facility, to Katja Beck and to the Data management group for excellent support with the next-generation sequencing analyses. Damir Krunic from the DKFZ Imaging core facility is acknowledged for his help with ImageJ macros. Marc Zuckermann and Emma Phillips are gratefully acknowledged for advice in the planning of the CRISPR experiments, Dominik Sturm for help with re-analysis of TCGA data, Susanne Gröbner for kindly helping with mutational signature analyses and Natalie Jäger for help with re-analysis of whole-genome sequencing data. We also thank Marius Wernig for kindly sharing the iPSC line 6-a (06C53141). We would like to express our special appreciation and thanks to Pierre-Olivier Frappart, Natalia Voronina, Kendra Maaß, Daisuke Kawauchi and Scott Pomeroy for discussions, and to Claus Bartram and Magnus von Knebel-Doeberitz for advice. We would like to thank Boston Children's Hospital Department of Medicine, the Charles H. Hood Foundation, and the Howard Hughes Medical Institute for supporting F.W.A laboratory and Charles A. King Trust Postdoctoral Research Fellowship Program, Bank of America, co-trustees for supporting P.-C.W. Finally, we would like to thank Michael Hain for excellent IT support, and the Fritz Thyssen Stiftung for financial support.

## Author contributions

M.R. performed most of the experiments; J.W., M.H., and Y.P. performed bioinformatic analyses; P.-C.W. performed the animal experiments with the Xrcc4/p53 mice; T.K. contributed to the CRISPR/Cas experiments; D.H. derived neural progenitors from human iPSCs; F.D. contributed to the FISH and sequencing experiments; R.K. provided the melanoma sequencing data; A.P. and W.M. contributed to the collection of human samples; A. K. performed neuropathology evaluation of tumor specimen; M.S. contributed to the cell culture experiments; A.J. analyzed the M-FISH data; D.T.W.J., M.K., P.N. contributed to data analysis; S.M.D. provided RNA-seq data; S.M.P., M.Z., P.J.M.K., F.W.A., P.L., and A.E. contributed to the original concept of the project and experimental design.

## Additional information

**Competing interests:** The authors declare no competing interests.

