## [Peer Review File · Nature Communications]

Reviewers' Comments:

Reviewer #1:

Remarks to the Author:

In this manuscript, the authors reported murine models of brain tumors that frequently showed catastrophic rearrangement events including chromothripsis and chromoanasythesis. These tumors, including medulloblastomas (MBs) or high-grade gliomas (HGGs), were derived from BRCA2/p53, XRCC4/p53 or LIG4/p53 inactivation. With the mouse model, the authors further investigated the roles of homologous recombination (HR) and classical non-homologous end-joining (c-NHEJ) in the generation of catastrophic events and the relationship between catastrophic rearrangements and gene amplification.

The murine models of catastrophic rearrangements created in this study are novel and useful tools for investigating the mechanisms of such events and their contributions to tumorigenesis. The analyses provided some new insight into the roles of different DNA repair mechanisms in generating complex rearrangements, including evidence supporting the involvement of alternative non-homologous end-joining when HR or c-NHEJ are inactive. The work is interesting overall, but its quality is undermined by several major technical issues, as discussed below. The authors should address these issues before the manuscript can be considered for publication.

Major comments:

(1) A major finding of this manuscript is the high prevalence of complex genomic rearrangements in MBs or HGGs in the absence of BRCA2, XRCC4 or LIG4 in p53-deficient mice. The authors concluded that most of these rearrangements are chromoanasythesis based on whole-genome sequencing analysis. But the bioinformatic analyses lack details to a degree that it is impossible to fully assess the validity of the results.

Complex rearrangement patterns (e.g., "chromoanasythesis" or "chromothripsis") are generally classified based on statistical criteria involving both DNA copy number and chromosomal rearrangements. The authors need to provide details on: a. how the DNA copy number is determined and how chromosomal rearrangements are detected; b. what are the criteria used to determine chromothripsis or chromoanasythesis; and importantly, whether the statistical criteria can clearly distinguish between chromothripsis and chromoanasythesis based on the observed events (DNA copy-number changes and rearrangements).

To address "a" the authors need to provide the relevant information on bioinformatic analysis, including both methods and statistics (detection sensitivity, etc.).

Addressing "b" is more complex. If the authors really want to claim a preponderance of chromoanasythesis, then they need to present a valid statistical framework to distinguish between chromoanasythesis and chromothripsis. We are skeptical that chromothripsis and chromoanasythesis can easily be distinguished from the genomic variants alone. We would therefore be satisfied if the authors stated that they observe complex rearrangements some of which are dominated by segmental gains and others dominated by losses.

In any case, to convincingly demonstrate chromothripsis or chromoanasythesis, the authors should show both global (e.g., CIRCOS plots) and blow-out views of DNA copy-number (or read depth coverage) and chromosomal rearrangements for every region inferred to have undergone chromothripsis or chromoanasythesis. Currently rearrangements were only shown for two cases (in Fig. 1b and Fig. 5b); DNA copy number is shown for all cases in Supp. Fig. 1 but there is no blow-out view of individual events.

The authors should also discuss issues including tumor purity, detection sensitivity, and detection specificity, and perform independent validation of (a subset of) rearrangements detected by

sequencing, by PCR or long-read sequencing.

Finally, we found several cases of chromoanasythesis inferred by the authors to be quite unconvincing. In Fig. 1b, the authors suggested that the Myc amplicon was part of chromoanasythesis on Chr. 15; but in Fig. 2b, the same Myc amplification was evidently contained in homologously staining regions (HSR); HSR is not generally thought to be related to chromoanasythesis but linked to double-minute chromosome integration or breakage-fusion-bridge cycles. Multiple examples of chromoanasythesis in Supp. Fig. 1 (such as Chr. 13 of MB1206 and Chr. 15 of MB1224) showed gradually increasing copy-number next to a sudden drop; such "cliff-like" patterns are often caused by breakage-fusion-bridge cycles. A careful examination of the DNA copy-number changes and chromosomal rearrangement patterns in these regions is necessary to determine whether these copy-number patterns truly reflect chromoanasythesis.

(2) The authors suggested that Myc/Mycn amplification may facilitate the generation of catastrophic events. This is an interesting proposal but the current study does not provide adequate evidence supporting this idea. The authors made the proposal based on three observations: a. Myc/Mycn amplification can occur independently from catastrophic events; b. Myc/Mycn amplification is clonal but catastrophic events are sometimes subclonal (thus Myc/Mycn amplification was inferred to be earlier than the catastrophic events); c. Myc/Mycn amplification is strongly linked with chromothripsis in human cancers. For a, the authors need to show the rearrangements in the Myc/Mycn amplicon and convincingly demonstrate that they are independent of the catastrophic events, some of which are adjacent to the Myc/Mycn amplicon (as suggested in (1)). For b, the observation of clonal Myc/Mycn amplification but subclonal presence of the derivative chromosome (remnant of the catastrophe) does not definitively prove that Myc amplification is an early event; it is possible that the catastrophic event had led to Myc amplification but the derivative chromosome does not offer any additional phenotypic advantage and was lost gradually during clonal expansion. For c, the association does not establish any causal relationship between Myc/Mycn amplification and chromothripsis. In summary, the authors can raise the interesting possibility that Myn/Mycn amplification drives the catastrophic events, but this should be presented as a hypothesis rather than as a firm conclusion.

(3) By generating tumors with chromothripsis in mice that are deficient in classical non-homologous end joining (Xrcc4/p53 or Lig4/p53), the authors demonstrated that classical NHEJ is not required for generating chromothripsis. By analyzing breakpoint junctions, the authors found out that in the absence of Xrcc4/p53 or Lig4/p53, the majority of breakpoints show 2-3 base pairs of homology, suggesting these junctions derive from alternative non-homologous end-joining. In contrast, in the absence of BRCA2, the breakpoint patterns are consistent with classic non-homologous end-joining repair. The authors should discuss and compare this result with another study (Ly et al., Nat Cell Biol 2016) that investigated the roles of homologous repair and non-homologous end-joining in chromothripsis, concluding that cNHEJ is the major joining pathway for chromothripsis (but the effects were small). Furthermore, although the shift towards longer junction homology is obvious in human SHH MBs (Fig. 5c), rearrangements in human glioblastomas (Supp. Fig. 10a) show the opposite trend (junction homology is shorter in chromothriptic rearrangements). The author should clarify the basis for this discrepancy.

(4) The authors also set out to investigate what leads to the catastrophic events in DNA repair deficient mice. In Figure 3 they report that in the absence of Xrcc4/p53, neural precursor cells exhibit a high level of DNA damage along with a defective apoptotic response. However, the differences in g-H2AX staining appear somewhat subtle and need to be quantified. (Also relevant to Sup. Fig. 4 & 5).

Minor comments:

Figure 1:

The y-axis labels in Fig.1a and Fig.1d are not consistent and confusing. The second plot in Fig.1a

did not clearly show complex gene rearrangement (Brca2 L/L p53 L/L Nestin-Cre (Mb270)-Chr. 13.) The data presented in Fig.1c should be presented as counts rather than percentage due to the small number of cases in each category.

Figure 2: The y-axis labels are not consistent in Fig.2b.

In Supplemental Figure 4, it is difficult to determine the quantitative differences in Ki67 between these samples by visual inspection. For example, it looks like the Ki67 is much weaker in the p53-/- Xrcc4c/- than p53-/- Xrcc4c/+ (P60 CBL). Representative images should be presented to support the conclusion in the text. In addition, the telomere FISH signals are difficult to interpret especially the P60 and P80 samples.

In Supplemental Figure 8, the TP53 knockout is clearly incomplete. We presume this is because the knockout was done in a population without obtaining clones. This should be clarified in the legend.

Reviewer #2:

Remarks to the Author:

Rathnaparke and colleagues performed whole-genome sequencing in tumours from various mouse models of medulloblastoma (MB) or high grade gliomas, with a focus on the characterization of complex clustered rearrangements. Their models are based on p53 inactivation in combination with inactivation of either homologous recombination repair (Brca2) or NHEJ (Xrcc4 or Lig4). The key findings are (i) frequent occurrence of catastrophic genomic events in mouse cancers (similarly to humans), (ii) the association of catastrophic events with Myc or Mycn gains (as in humans), (iii) insights into the repair processes active at the time of the catastrophic event, (iv) analyses on therapeutic exploitability of DNA repair defects in tumours with complex genomic rearrangements. This is one of the first studies performing such comprehensive (whole-genome sequencing based) analyses of mouse cancers. Given the growing importance of mouse models in cancer research, it is essential to understand how well mouse cancer genetics reflects the human situation. Thus, the study - performed by leading groups in the field - is in my eyes interesting and important. Some of the data also have immediate translational relevance.

However, the manuscript would benefit from some more focus. It touches on many different aspects, and in several cases the conclusions need some more experimental evidence. The authors should consider (i) to remove some of the speculative findings which are not in the focus of the manuscript, and (ii) in turn to provide some more data to support the key aspects. Some points need clarification.

Specific comments:

1. The manuscript would benefit from a comprehensive presentation of genomic data emanating from the different mouse models. I would suggest to present all analyses, including SNV, CNV etc and to compare their frequency to similar human cancers. For various entities, mouse cancers are typically having fewer SNVs than their human counterparts. Is this also true for brain cancers analysed?

2. The models and cross-species comparisons. It seems that the models were chosen because tumours in mice having compound inactivation of p53 and different repair pathway components frequently display catastrophic genomic events. However, tumours without repair pathway deficiencies were not analysed and it remains unclear whether the development of these catastrophic events is indeed linked to the repair pathway deficiencies or whether it merely reflects cell-type specific propensity for the acquisition of these catastrophic rearrangements, as observed for other cancer types. Please comment. Also the cross-species comparisons need to be better explained, as full (homozygous) inactivation of repair pathway components seems to be rare in published human data on medulloblastoma and glioma. The related data in supplementary figure 10b need to be better explained. Furthermore, is it possible, that the differences between mouse

(more chromoanasythesis) and man (more chromotripsis) are linked to the specific models? What is the rate of chromoanasythesis in human MBs and gliomas with homozygous inactivation of DNA repair pathway genes?

3. Several parts of the manuscript might benefit from provision of some more methodological details. For example:

- It should be described in more detail how complex/clustered rearrangements were defined and scored. For instance: (i) What are the criteria for distinguishing chromothripsis from chromoanasythesis and how were the tumours scored? A list of scoring criteria should be provided and it should be stated for each individual tumour whether criteria are applying or not.

- Page 4; 2nd paragraph: In the last sentence of this paragraph it is stated that chromosomes affected by chromoanasythesis were also shown to harbour recurrent breakpoints. How was this analysed and at which position of the manuscript are the results shown?

- Page 4, last paragraph: It is stated that Mycn and/or Myc loci are occasionally included in the region of complex rearrangements. The authors might consider to extend supplementary table1 with the exact chromosomal regions affected by the complex rearrangement and the genomic position of the amplified Mycn/Myc segment.

4. Page 5, 1st and 2nd paragraph: The authors might consider to put less emphasis on discussing the sequential order of Myc/Mycn and complex rearrangements, as (i) the evidence of Myc amplification preceding complex rearrangements is based on only two mouse tumours and (ii) because they show (in suppl. figure 9) that in some cancers MYC is in fact the most significant gene within chromotriptic regions. The latter suggests that chromotripsis being first and MYC gain being selected for subsequently (at least in some cancers). So it seems that both scenarios are occurring. Regarding (i), more data would need to be provided to strengthen the conclusion. For example, for each MB cell line (594 and 794) both clones should be analysed separately. This would allow to test if both clones from a cell line show the same Mycn/Myc breakpoint junctions. Identical breakpoints would prove that both clones originate from the same ancestor.

5. One of the most important finding of this manuscript is the discovery of potential therapeutic vulnerabilities in mouse cancers with complex rearrangements. The study shows efficacy of HR and/or Alt-EJ inhibitors in combination with topotecan in mouse brain tumours with DNA repair deficiencies. It also demonstrates that corresponding human brain tumours with BRCA2 deficiency are sensitive to such treatment. However, it remains unclear whether this sensitivity applies generally to the large number of chromotriptic human brain tumours, as only a small part of them seems to have homozygous inactivation of DNA repair genes (although the breakpoint analyses suggest some level of deficiency!). This could be clarified by analysing human MB and glioma cancer cell lines with chromotripsis. The results would further increase the impact of the study.

Reviewer #3:

Remarks to the Author:

The importance in cancer cells of chromothripsis and chromoanasythesis, processes for which one or several chromosomes are broken into many fragments subsequently joined in random order and orientation, has been appreciated only recently. The biochemical and signaling mechanisms underlying these complex genetic rearrangements still remain to be identified.

In this manuscript Ratnaparke and colleagues investigate the role of different DNA repair factors in chromothripsis and chromoanasythesis in brain tumors.

By using whole genome sequencing the authors observed catastrophic genetic rearrangements in different mouse models defective for either homologous-recombination-repair (BRCA2/p53 deficient) or nonhomologous-end-joining (XRCC4/p53 and LIG4/p53 deficient). Furthermore, they detected frequent amplification in a variety of oncogenes, such as Myc and Mycn.

The authors suggest that the defects in DNA damage repairs contribute/lead to catastrophic genomic events associated to chromothripsis, possibly facilitated by the amplification of Myc and Mycn. Moreover, they propose that those alterations could represent vulnerabilities that could be

exploited from a therapeutic stand point.

The study is novel and potentially interesting, however, some of the authors' conclusion are quite speculative and not extensively supported by the data presented. One of the possible major limitation, partially highlighted also by the authors in the discussion section, is that the experimental setting used in this manuscript is not sufficient to "distinguish between the contribution of p53 and the role of the repair factors themselves". The role of p53 in chromothripsis has been described previously and in the current study it's not obvious whether defects in the DNA repair further exacerbate it. Although, Ratnaparke and colleagues present (known/expected) evidence of increased DNA damage in the DNA repair defective background, how this contribute to chromothripsis is not sufficiently addressed. To better support their hypothesis for a role of defects in HR and NHEJ in chromothripsis, the authors should present either an in vivo model or a cellular system (possibly the described CRISPR/Cas9 in iPSC cells or the cellular model published in Mardin et al. Mol Syst Biol, 2015) to direct compare DNA repair defects alone or in combination with p53 deficiency.

Moreover, the putative driver role of MYC in the catastrophic genetic events observed in the different tumor model and human samples appears to be only correlative and it is not addressed experimentally. It is conceivable that high level of Myc expression are required for proliferation in cells that show chromothripsis. To claim causality, the authors should present an experimental model in which they overexpress Myc, otherwise they should be more conservative in the results and discussion section when they describe the amplification of MYC in the samples with chromothripsis.

Further comments:

- The authors refer to the Methods section on how they score for the chromothripsis and chromoanasythesis events. For the sake of clarity, they should briefly describe it also in the first paragraph of the results section.
- Figure S1, please specify what are the red boxes highlighting.
- Figure 3, the authors should present quantification of the pH2AX and TUNEL staining.
- Supplementary table, please correct the title of Suppl Table 1 in the Excel file.
- The CRISPR/Cas9 editing in iPSC cells should be better described. Do the authors use a mixed cell population or did they isolate single clones? Why are the cells indicated as LIG4 +/- and not -/-?
- Supplementary Figure 10b, what are the levels of expression of the DNA repair genes in the samples with hemizygous deletion? Hemizigosity is not always associated with lower mRNA expression, there might be compensatory mechanism that increase their expression from the remaining allele.
- There is a general lack of statistical tests throughout the manuscript. Most of the figures and legends should be revised to specify what test statistical tests have been used to claim significant differences. Similarly, the data regarding MYC amplification presented in supplementary figure 2 should supported by statistical analysis.

Reviewer #4:

Remarks to the Author:

The authors tried to illustrate that defection of DNA damage repair genes can frequently lead to

catastrophic genomic events in mice and human tumors. The finding is interesting, but the presentation of the current manuscript prevents readers to fully follow the authors' logic. My major concerns are listed as follows.

1, The authors did not clearly demonstrate the experimental design of the whole study that is tightly associated to their claims. In the Online Methods section, only details of sequencing and validation were provided. The types, groups, numbers of experimental mice were totally missing. Without these details, readers neither can understand how the authors' conclusions were generated nor will reproduce their findings.

2, The data processing details were incomplete. The authors did not provide sufficient details of the bioinformatics analyses but only provided a name of the software without references.

3, The determination of chromothripsis and chromoanasythesis, two core concepts of this manuscript, were completely unstated. How did the authors quantify and discriminate chromothripsis and chromoanasythesis based on the sequencing data?

4, Statistical analyses were also missing in the current manuscript. Although the numbers of positive/total mice were provided in the Results section, it is hard to assess the significance of the observations.

5, Many statements were only descriptive or even speculative. For example, "Cases with several normal copies and one larger derivative chromosome (presumably the copy affected by chromoanasythesis) suggest that polyploidization may precede chromoanasythesis, facilitating the survival of the cell after such a catastrophic event" provides no information but may mislead readers because the authors neither clearly stated the phenomenon nor clarified all the behind possibilities.

Reviewers' comments:

Reviewer #1 (Remarks to the Author):

In this manuscript, the authors reported murine models of brain tumors that frequently showed catastrophic rearrangement events including chromothripsis and chromoanasythesis. These tumors, including medulloblastomas (MBs) or high-grade gliomas (HGGs), were derived from BRCA2/p53, XRCC4/p53 or LIG4/p53 inactivation. With the mouse model, the authors further investigated the roles of homologous recombination (HR) and classical non-homologous end-joining (c-NHEJ) in the generation of catastrophic events and the relationship between catastrophic rearrangements and gene amplification.

The murine models of catastrophic rearrangements created in this study are novel and useful tools for investigating the mechanisms of such events and their contributions to tumorigenesis. The analyses provided some new insight into the roles of different DNA repair mechanisms in generating complex rearrangements, including evidence supporting the involvement of alternative non-homologous end-joining when HR or c-NHEJ are inactive. The work is interesting overall, but its quality is undermined by several major technical issues, as discussed below. The authors should address these issues before the manuscript can be considered for publication.

Major comments:

(1) A major finding of this manuscript is the high prevalence of complex genomic rearrangements in MBs or HGGs in the absence of BRCA2, XRCC4 or LIG4 in p53-deficient mice. The authors concluded that most of these rearrangements are chromoanasythesis based on whole-genome sequencing analysis. But the bioinformatic analyses lack details to a degree that it is impossible to fully assess the validity of the results.

Author's response:

In the revised version of the manuscript, we added details about the bioinformatic analyses to the methods (see pages 15 to 18). We also used a new pipeline to visualize the complex rearrangements and better assess the different scoring criteria (e.g. arcs to show the rearrangements, dotted lines to indicate the segment counts...) and ran an algorithm to detect complex rearrangements from WGS data (ShatterSeek, developed by Park et al, <https://github.com/parklab/ShatterSeek> , Akbani et al., 2015, Cell).

Complex rearrangement patterns (e.g., "chromoanasythesis" or "chromothripsis") are generally classified based on statistical criteria involving both DNA copy number and chromosomal rearrangements. The authors need to provide details on:

- a. how the DNA copy number is determined and how chromosomal rearrangements are detected;
- b. what are the criteria used to determine chromothripsis or chromoanasythesis; and importantly, whether the statistical criteria can clearly distinguish between chromothripsis and chromoanasythesis based on the observed events (DNA copy-number changes and rearrangements).

To address "a" the authors need to provide the relevant information on bioinformatic analysis, including both methods and statistics (detection sensitivity, etc.).

Addressing "b" is more complex. If the authors really want to claim a preponderance of chromoanasythesis, then they need to present a valid statistical framework to distinguish between chromoanasythesis and chromothripsis. We are skeptical that chromothripsis and chromoanasythesis can easily be distinguished from the genomic variants alone.

We would therefore be satisfied if the authors stated that they observe complex rearrangements some of which are dominated by segmental gains and others dominated by losses.

Author's response:

We agree that it may be challenging to provide a statistical framework to distinguish between chromoanasythesis and chromothripsis.

As suggested, we now refer to the observed catastrophic events as complex rearrangements dominated by segmental gains or dominated by losses, respectively.

In any case, to convincingly demonstrate chromothripsis or chromoanasythesis, the authors should show both global (e.g., CIRCOS plots) and blow-out views of DNA copy-number (or read depth coverage) and chromosomal rearrangements for every region inferred to have undergone chromothripsis or chromoanasythesis. Currently rearrangements were only shown for two cases (in Fig. 1b and Fig. 5b); DNA copy number is shown for all cases in Supp. Fig. 1 but there is no blow-out view of individual events.

Author's response:

As requested, we now added to Supplementary figure 1 all CIRCOS plots and blow-out views of the chromosomes affected by catastrophic events. In addition, we added a Supplementary Table (Suppl Table 2) with the output of the Shatterseek algorithm developed to detect complex rearrangements.

The authors should also discuss issues including tumor purity, detection sensitivity, and detection specificity, and perform independent validation of (a subset of) rearrangements detected by sequencing, by PCR or long-read sequencing.

Author's response:

We now added the information on tumor cell content based on neuropathological evaluation of hematoxylin and eosin stainings of the tumors in the Methods section (page 12, paragraph 1). We also included the calculations of the detection sensitivity (see page 1g, paragraph 3).

As suggested, we performed an independent validation of a subset of rearrangements by PCR. Twelve out of 14 breakpoints were validated, the results are shown in Supplementary Figure 1 (last panel).

Finally, we found several cases of chromoanasythesis inferred by the authors to be quite unconvincing. In Fig. 1b, the authors suggested that the *Myc* amplicon was part of chromoanasythesis on Chr. 15; but in Fig. 2b, the same *Myc* amplification was evidently contained in homologically staining regions (HSR); HSR is not generally thought to be related to chromoanasythesis but linked to double-minute chromosome integration or breakage-fusion-bridge cycles. Multiple examples of chromoanasythesis in Supp. Fig. 1 (such as Chr. 13 of MB1206 and Chr. 15 of MB1224) showed gradually increasing copy-number next to a sudden drop; such "cliff-like" patterns are often caused by breakage-fusion-bridge cycles. A careful examination of the DNA copy-number changes and chromosomal rearrangement patterns in these regions is necessary to determine whether these copy-number patterns truly reflect chromoanasythesis.

To provide better evidence of the scoring criteria, we now included blow-out views of the affected chromosomes, arcs to show the rearrangements, Shatterseek scoring in addition to the visual inspection (see Supplementary Figure 1 and Supplementary Table 2).

We agree that HSR with *Myc* amplifications as for MB794 may be linked to double-minute integration or breakage-fusion-bridge cycles and that the "click-off" patterns as seen in several cases may be due to breakage-fusion-bridge cycles. However, both double-minute chromosomes and breakage-fusion-bridge cycles have been linked to catastrophic events previously (Rausch et al, Cell 2012; Li, Campbell, Harrison et al, Nature 2014; Maciejowski, Campbell, De Lange et al, Cell 2015). Therefore, we do not see any contradiction between the detection of catastrophic events and HSR or scars of breakage-fusion-bridge cycles.

(2) The authors suggested that *Myc/Mycn* amplification may facilitate the generation of catastrophic events. This is an interesting proposal but the current study does not provide adequate evidence supporting this idea. The authors made the proposal based on three observations:

- a. *Myc/Mycn* amplification can occur independently from catastrophic events;
- b. *Myc/Mycn* amplification is clonal but catastrophic events are sometimes subclonal (thus *Myc/Mycn* amplification was inferred to be earlier than the catastrophic events);
- c. *Myc/Mycn* amplification is strongly linked with chromothripsis in human cancers.

For a, the authors need to show the rearrangements in the *Myc/Mycn* amplicon and convincingly demonstrate that they are independent of the catastrophic events, some of which are adjacent to the *Myc/Mycn* amplicon (as suggested in (1)).

For b, the observation of clonal *Myc/Mycn* amplification but subclonal presence of the derivative chromosome (remnant of the catastrophe) does not definitively prove that *Myc* amplification is an early event; it is possible that the catastrophic event had led to *Myc* amplification but the derivative chromosome does not offer any additional phenotypic advantage and was lost gradually during clonal expansion.

For c, the association does not establish any causal relationship between *Myc/Mycn* amplification and chromothripsis.

In summary, the authors can raise the interesting possibility that *Myc/Mycn* amplification drives the catastrophic events, but this should be presented as a hypothesis rather than as a firm conclusion.

As suggested, we now mention the possibility that *Myc/Mycn* amplification might drive the catastrophic events and not necessarily be a selective advantage resulting from catastrophic events as a hypothesis only (see text page 5). We removed the related data from the main figure and now show these results in Supplementary Figure 3.

We agree that the catastrophic event can lead to the formation of a derivative chromosome and double-minute chromosomes carrying *Myc/Mycn*, with subsequent loss of the derivative chromosome and retention of the double-minute chromosome; we added this possibility to the text (see page 6, paragraph 1).

(3) By generating tumors with chromothripsis in mice that are deficient in classical non-homologous end joining (*Xrcc4/p53* or *Lig4/p53*), the authors demonstrated that classical NHEJ is not required for generating chromothripsis. By analyzing breakpoint junctions, the authors found out that in the absence of *Xrcc4/p53* or *Lig4/p53*, the majority of breakpoints show 2-3 base pairs of homology, suggesting these junctions derive from alternative non-homologous end-joining. In contrast, in the absence of *BRCA2*, the breakpoint patterns are consistent with classic non-homologous end-joining repair. The authors should discuss and compare this result with another study (Ly et al., *Nat Cell Biol* 2016) that investigated the roles of homologous repair and non-homologous end-joining in chromothripsis, concluding that cNHEJ is the major joining pathway for chromothripsis (but the effects were small).

As suggested, we now added a section in the discussion to compare our results with the data from Ly et al (page 10, paragraph 2).

Furthermore, although the shift towards longer junction homology is obvious in human SHH MBs (Fig. 5c), rearrangements in human glioblastomas (Supp. Fig. 10a) show the opposite trend (junction homology is shorter in chromothriptic rearrangements). The author should clarify the basis for this discrepancy.

As this shift towards longer junction homology is highly significant in SHH MBs but the analysis of the microhomologies at the breakpoints did not show any significant difference for the glioblastomas (borderline of significance for 0-1 bp and not significant for 2-3 bp between regions affected by complex genome rearrangements and cases without), we refrained from pointing out the results of the breakpoint analysis in the glioblastomas.

However, it is conceivable that pronounced differences across entities may exist in terms of repair processes, and that distinct mechanisms of genome instability may be active in different tumor entities, with different processes leading to highly rearranged derivative chromosomes (Gröbner et al, Nature 2018).

(4) The authors also set out to investigate what leads to the catastrophic events in DNA repair deficient mice. In Figure 3 they report that in the absence of Xrcc4/p53, neural precursor cells exhibit a high level of DNA damage along with a defective apoptotic response. However, the differences in g-H2AX staining appear somewhat subtle and need to be quantified. (Also relevant to Sup. Fig. 4 & 5).

As requested, we now added the quantifications to Figure 3 by counting the number of positive cells (details are provided in the Methods section), as well as for Supplementary Figures 4 and 5.

Minor comments:

Figure 1:

The y-axis labels in Fig.1a and Fig.1d are not consistent and confusing.

As suggested, we now changed the y-axis labels to make them consistent.

The second plot in Fig.1a did not clearly show complex gene rearrangement (Brca2 L/L p53 L/L Nestin-Cre (Mb270)-Chr. 13.)

We now used a different pipeline and replaced all the plots to visualize the complex rearrangements more clearly.

The data presented in Fig.1c should be presented as counts rather than percentage due to the small number of cases in each category.

As suggested, we added the counts in Figure 1c on the top of the bars for each genotype.

Figure 2: The y-axis labels are not consistent in Fig.2b.

As suggested, we changed the y-axis labels to make them consistent.

The results previously shown in Figure 2b are now included in Supplementary Figure 3.

In Supplemental Figure 4, it is difficult to determine the quantitative differences in Ki67 between these samples by visual inspection. For example, it looks like the Ki67 is much weaker in the p53^{-/-} Xrcc4^{-/-} than p53^{-/-} Xrcc4^{+/+} (P60 CBL). Representative images should be presented to support the conclusion in the text. In addition, the telomere FISH signals are difficult to interpret especially the P60 and P80 samples.

We added quantifications to Supplementary Figure 4 by counting the number of positive cells (details are provided in the Methods section). We selected images that are more representative of the main findings for the Ki67 stainings and for the telomere FISH. Furthermore, we added quantifications of the telomere length based on the analysis of the telomeric reads from the whole-genome sequencing data to provide an independent assessment of the telomere length.

In Supplemental Figure 8, the TP53 knockout is clearly incomplete. We presume this is because the knockout was done in a population without obtaining clones. This should be clarified in the legend.

As requested, we provided more details on the CRISPR/Cas editing in the methods (page 14) and in the legend regarding the *TP53* knockout with comprehensive information about the different steps (e.g. selection of the CRISPR-edited cells via puromycin and/or blasticidin treatment followed by nutlin treatment to validate the functional knock-out...).

In addition, we repeated the experiment with new guide RNAs and obtained similar results after complete knockout also visible on the western blots (see Figure 5 and Supplementary Figure 8).

Reviewer #2 (Remarks to the Author):

Rathnaparke and colleagues performed whole-genome sequencing in tumours from various mouse models of medulloblastoma (MB) or high grade gliomas, with a focus on the characterization of complex clustered rearrangements. Their models are based on p53 inactivation in combination with inactivation of either homologous recombination repair (Brca2) or NHEJ (Xrcc4 or Lig4). The key findings are (i) frequent occurrence of catastrophic genomic events in mouse cancers (similarly to humans), (ii) the association of catastrophic events with Myc or Mycn gains (as in humans), (iii) insights into the repair processes active at the time of the catastrophic event, (iv) analyses on therapeutic exploitability of DNA repair defects in tumours with complex genomic rearrangements.

This is one of the first studies performing such comprehensive (whole-genome sequencing based) analyses of mouse cancers. Given the growing importance of mouse models in cancer research, it is essential to understand how well mouse cancer genetics reflects the human situation. Thus, the study - performed by leading groups in the field - is in my eyes interesting and important. Some of the data also have immediate translational relevance. However, the manuscript would benefit from some more focus. It touches on many different aspects, and in several cases the conclusions need some more experimental evidence. The authors should consider (i) to remove some of the speculative findings which are not in the focus of the manuscript, and (ii) in turn to provide some more data to support the key aspects. Some points need clarification.

Specific comments:

1. The manuscript would benefit from a comprehensive presentation of genomic data emanating from the different mouse models. I would suggest to present all analyses, including SNV, CNV etc and to compare their frequency to similar human cancers.

As suggested, we added a comparison of the genomic data to human medulloblastomas and high-grade gliomas (new panel f in Figure 1 including SNVs, Indels, and structural variants).

For various entities, mouse cancers are typically having fewer SNVs than their human counterparts. Is this also true for brain cancers analysed?

Yes, the mouse models of medulloblastoma and high-grade glioma analyzed in this study show significantly less SNVs as compared to human medulloblastoma and high-grade gliomas, respectively. We added a sentence to mention this point in the results section (see page 5, paragraph 2).

2. The models and cross-species comparisons. It seems that the models were chosen because tumours in mice having compound inactivation of p53 and different repair pathway components frequently display catastrophic genomic events. However, tumours without repair pathway deficiencies were not analysed and it remains unclear whether the development of these catastrophic events is indeed linked to the repair pathway deficiencies or whether it merely reflects cell-type specific propensity for the acquisition of these catastrophic rearrangements, as observed for other cancer types. Please comment.

We now performed chromothripsis scoring for murine models of medulloblastoma not based on repair pathway deficiencies, for which whole genome sequencing data were available

from two previously published studies (Pöschl, Stark et al, Acta Neuropathol 2014 and Akgül et al, Cell Reports 2018).

The results are presented in the new Supplementary figure 2.

We also commented on this point in the discussion (page 11, last sentence of the first paragraph).

Furthermore, we added a number of controls to the mutational signature analysis (Figure 4d, Supplementary Figure 6).

As we had detected mutational signature 3 (linked with homologous recombination deficiency) in murine tumors that are homologous recombination proficient, we suspected that mutational signature 3 might be linked to further defects (beyond HR deficiency), including for instance p53 deficiency. We performed mutational signature analysis in published mouse models with and without p53 deficiency, respectively, to evaluate this possibility.

Also the cross-species comparisons need to be better explained, as full (homozygous) inactivation of repair pathway components seems to be rare in published human data on medulloblastoma and glioma. The related data in supplementary figure 10b need to be better explained.

Furthermore, is it possible, that the differences between mouse (more chromoanasythesis) and man (more chromotripsis) are linked to the specific models?

What is the rate of chromoanasythesis in human MBs and gliomas with homozygous inactivation of DNA repair pathway genes?

As suggested, we provided more details in the legend of Supplementary Figure 10b and also added the expression values for the respective factors (new panel in Supplementary Figure 10b).

We agree that it is conceivable that the differences between mouse and humans in terms of prevalence for chromotripsis and chromoanasythesis may be linked to specific models and added this possibility to the discussion (page 11).

Homozygous inactivation of repair pathway components is very rare in humans, as mentioned by Reviewer #2. We added a figure (Figure 5d) and a table (Supplementary table 4) on the prevalence of catastrophic events in human MBs and HGGs with inactivation of DNA repair pathway genes. For this, we scored catastrophic events based on whole-genome sequencing data from two published human studies (Gröbner et al, Nature 2018 and Waszak & Northcott et al, Lancet Oncology 2018).

3. Several parts of the manuscript might benefit from provision of some more methodological details. For example:

- It should be described in more detail how complex/clustered rearrangements were defined and scored. For instance: (i) What are the criteria for distinguishing chromotripsis from chromoanasythesis and how were the tumours scored? A list of scoring criteria should be provided and it should be stated for each individual tumour whether criteria are applying or not.

As requested, we now provided more details about the bioinformatics methods (see page...), and in particular the scoring criteria for catastrophic events. This point was also raised by Reviewer #1 (point 1). As suggested by Reviewer #1, we now use the term catastrophic events “dominated by gains” or “dominated by losses” as it is questionable whether a statistical framework could allow distinguishing between both types of events.

- Page 4; 2nd paragraph: In the last sentence of this paragraph it is stated that chromosomes

affected by chromoanasythesis were also shown to harbour recurrent breakpoints. How was this analysed and at which position of the manuscript are the results shown?

This sentence refers to a study by Yan, Alt et al (PNAS 2006, citation in the end of the sentence). This mouse model was originally published in the study by Yan, Alt et al and at that time CGH and FISH analyses were performed. The chromosomes shown to be affected by recurrent breakpoints in the study by Yan, Alt and colleagues are shown to be affected by catastrophic events in the new (independent) cohort of mice based on our WGS data. We now rephrased this sentence to clarify this point.

- Page 4, last paragraph: It is stated that *Mycn* and/or *Myc* loci are occasionally included in the region of complex rearrangements. The authors might consider to extend supplementary table1 with the exact chromosomal regions affected by the complex rearrangement and the genomic position of the amplified *Mycn/Myc* segment.

As suggested, the genomic positions of the amplified *Mycn/Myc* loci were added to Supplementary Table 1 and to Table 1, and we also added the information whether these loci were included in the regions affected by complex rearrangements. The exact chromosomal regions affected by complex rearrangements are also highlighted by red boxes on the coverage plots in Supplementary Figure 1 and indicated in Supplementary Table 2.

4. Page 5, 1st and 2nd paragraph: The authors might consider to put less emphasis on discussing the sequential order of *Myc/Mycn* and complex rearrangements, as

(i) the evidence of *Myc* amplification preceding complex rearrangements is based on only two mouse tumours and

(ii) because they show (in suppl. figure 9) that in some cancers *MYC* is in fact the most significant gene within chromotriptic regions. The latter suggests that chromotripsis being first and *MYC* gain being selected for subsequently (at least in some cancers). So it seems that both scenarios are occurring. Regarding (i), more data would need to be provided to strengthen the conclusion. For example, for each MB cell line (594 and 794) both clones should be analysed separately.

This would allow to test if both clones from a cell line show the same *Mycn/Myc* breakpoint junctions. Identical breakpoints would prove that both clones originate from the same ancestor.

As suggested, we now put less emphasis on the serial order of *Myc/Mycn* gains and complex rearrangements. The related data were moved from Figure 2b to Supplementary Figure 3 and we revised the text accordingly (see page 5).

This point was also raised by Reviewers #1 and #3, and we now performed chromotripsis scoring for murine models of medulloblastoma based on *Myc* overexpression, for which whole genome sequencing data were available from a previously published study (Pöschl, Stark et al, Acta Neuropathol 2014). The results are presented in the new Supplementary figure 2b. In these *MYC*-driven models, only 1 out of 15 medulloblastomas showed a catastrophic event, showing that *Myc/Mycn* overexpression in neural progenitors in the absence of DNA repair defects does not facilitate chromotripsis. Therefore, we now mention the possibility of a causative role for *Myc/Mycn* in a more conservative way, as the role of *MYC/MYCN* is highly context dependent.

5. One of the most important finding of this manuscript is the discovery of potential therapeutic vulnerabilities in mouse cancers with complex rearrangements. The study shows efficacy of HR and/or Alt-EJ inhibitors in combination with topotecan in mouse brain tumours with DNA repair deficiencies. It also demonstrates that corresponding human brain tumours with *BRCA2* deficiency are sensitive to such treatment. However, it remains unclear whether this sensitivity applies generally to the large number of chromotriptic human brain tumours, as only a small part of them seems to have homozygous inactivation of DNA repair genes

(although the breakpoint analyses suggest some level of deficiency!). This could be clarified by analysing human MB and glioma cancer cell lines with chromotripsis. The results would further increase the impact of the study.

As suggested, we applied PARPi in combination with topoisomerase inhibitors on brain tumor cell lines with chromotripsis (one MB and one high-grade glioma line). Both lines were sensitive to the treatment; the results are shown in Figure 4f.

Reviewer #3 (Remarks to the Author):

The importance in cancer cells of chromotripsis and chromoanasythesis, processes for which one or several chromosomes are broken into many fragments subsequently joined in random order and orientation, has been appreciated only recently. The biochemical and signaling mechanisms underlying these complex genetic rearrangements still remain to be identified.

In this manuscript Ratnaparke and colleagues investigate the role of different DNA repair factors in chromotripsis and chromoanasythesis in brain tumors.

By using whole genome sequencing the authors observed catastrophic genetic rearrangements in different mouse models defective for either homologous-recombination-repair (BRCA2/p53 deficient) or nonhomologous-end-joining (XRCC4/p53 and LIG4/p53 deficient). Furthermore, they detected frequent amplification in a variety of oncogenes, such as Myc and Mycn. The authors suggest that the defects in DNA damage repairs contribute/lead to catastrophic genomic events associated to chromotripsis, possibly facilitated by the amplification of Myc and Mycn. Moreover, they propose that those alterations could represent vulnerabilities that could be exploited from a therapeutic stand point.

The study is novel and potentially interesting, however, some of the authors' conclusion are quite speculative and not extensively supported by the data presented. One of the possible major limitation, partially highlighted also by the authors in the discussion section, is that the experimental setting used in this manuscript is not sufficient to "distinguish between the contribution of p53 and the role of the repair factors themselves". The role of p53 in chromotripsis has been described previously and in the current study it's not obvious whether defects in the DNA repair further exacerbate it.

Although, Ratnaparke and colleagues present (known/expected) evidence of increased DNA damage in the DNA repair defective background, how this contribute to chromotripsis is not sufficiently addressed. To better support their hypothesis for a role of defects in HR and NHEJ in chromotripsis, the authors should present either an *in vivo* model or a cellular system (possibly the described CRISPR/Cas9 in iPSC cells or the cellular model published in Mardin et al. Mol Syst Biol, 2015) to direct compare DNA repair defects alone or in combination with p53 deficiency.

We agree that a direct comparison of the DNA repair defects alone or in combination with p53 deficiency, ideally in an *in vivo* model would be very important. Unfortunately the single knock-out for Xrcc4 is embryonic lethal (Yan, Alt et al, PNAS 2006) and the single knock-out for p53 does not lead to medulloblastoma development, therefore a direct comparison of the chromotripsis prevalence in the exact same background is not possible.

However, we performed a chromotripsis scoring for murine models of medulloblastoma based on p53 deficiency in combination with activation or inactivation of various oncogenes and tumor suppressor genes without any direct involvement of DNA repair, for which whole genome sequencing data were available from two previously published studies (Pöschl, Stark et al, Acta Neuropathol 2014 and Akgül et al, Cell Reports 2018). The results are presented in the new Supplementary figure 2b.

Based on the chromothripsis prevalence in these models, p53 inactivation alone is not sufficient in murine medulloblastomas to lead to catastrophic events.

Moreover, the putative driver role of MYC in the catastrophic genetic events observed in the different tumor model and human samples appears to be only correlative and it is not addressed experimentally. It is conceivable that high level of Myc expression are required for proliferation in cells that show chromothripsis. To claim causality, the authors should present an experimental model in which they overexpress Myc, otherwise they should be more conservative in the results and discussion section when they describe the amplification of MYC in the samples with chromothripsis.

This point was also raised by Reviewer #2. We now performed scoring for complex genome rearrangements for murine models of medulloblastoma based on *Myc/Mycn* overexpression, for which whole genome sequencing data were available from a previously published study (Pöschl, Stark et al, Acta Neuropathol 2014). The results are presented in the new supplementary figure 2b. In these MYC-driven models (also in a p53-deficient background), only 1 out of 15 medulloblastomas showed a catastrophic event, showing that *Myc/Mycn* overexpression in neural progenitors in the absence of DNA repair defects does not facilitate chromothripsis. Therefore, we now mention the possibility of a causative role for *Myc/Mycn* in a more conservative way and removed these results from the main figure.

Further comments:

- The authors refer to the Methods section on how they score for the chromothripsis and chromoanasythesis events. For the sake of clarity, they should briefly describe it also in the first paragraph of the results section.

As suggested, we now added a brief description of the scoring criteria for catastrophic events in the results (page 4, first paragraph).

- Figure S1, please specify what are the red boxes highlighting.

The red boxes mark the chromosomes affected by catastrophic events. We now added this information to the legend of Supplementary Figure 1.

- Figure 3, the authors should present quantification of the pH2AX and TUNEL staining.

As requested, we now quantified the pH2AX and TUNEL stainings by counting the number of positive cells (details are provided in the Methods section) and included these data in the revised Figure 3.

- Supplementary table, please correct the title of Suppl Table 1 in the Excel file.

In the revised version of the manuscript, the title of Suppl Table 1 was corrected.

- The CRISPR/Cas9 editing in iPSC cells should be better described. Do the authors use a mixed cell population or did they isolate single clones? Why are the cells indicated as LIG4 +/- and not -/-?

As requested, we added details to the experimental procedures on the CRISPR/Cas9 editing (see Methods, page 14). We used a mixed cell population. We now performed additional experiments with new gRNAs and showed the successful gene disruption (see new Supplementary Figure 8). We also corrected the naming of the CRISPR-edited cells.

- Supplementary Figure 10b, what are the levels of expression of the DNA repair genes in the samples with hemizygous deletion? Hemizigosity is not always associated with lower mRNA expression, there might be compensatory mechanism that increase their expression from the remaining allele.

As suggested, we now added the expression levels of the DNA repair genes in Supplementary Figure 10b as well as the significance analysis for differences in expression levels.

- There is a general lack of statistical tests throughout the manuscript. Most of the figures and legends should be revised to specify what statistical tests have been used to claim significant differences. Similarly, the data regarding MYC amplification presented in supplementary figure 2 should be supported by statistical analysis.

As requested, statistical analyses were added on revised Figure 4 and on Supplementary Figures 2 and 6. The new legends now include the names of the statistical tests used.

Reviewer #4 (Remarks to the Author):

The authors tried to illustrate that defection of DNA damage repair genes can frequently lead to catastrophic genomic events in mice and human tumors. The finding is interesting, but the presentation of the current manuscript prevents readers to fully follow the authors' logic. My major concerns are listed as follows.

1, The authors did not clearly demonstrate the experimental design of the whole study that is tightly associated to their claims. In the Online Methods section, only details of sequencing and validation were provided. The types, groups, numbers of experimental mice were totally missing. Without these details, readers neither can understand how the authors' conclusions were generated nor will reproduce their findings.

In the new version of the manuscript, all details on genotypes, groups and numbers of animals are provided.

2, The data processing details were incomplete. The authors did not provide sufficient details of the bioinformatics analyses but only provided a name of the software without references. As requested, we now added more details on the bioinformatics analyses, including references (see page 15 to 18).

3, The determination of chromothripsis and chromoanasythesis, two core concepts of this manuscript, were completely unstated. How did the authors quantify and discriminate chromothripsis and chromoanasythesis based on the sequencing data?

This point was also raised by Reviewers # 1 and 2.

As requested, we added details about the bioinformatic analyses to the methods and specifically about the scoring criteria (see page 17). We also used a new pipeline to visualize the rearrangements and better assess the different criteria (e.g. arcs to show the rearrangements, dotted lines to display the segment counts...) and ran an algorithm to detect complex rearrangements from WGS data (ShatterSeek, developed by Park et al, <https://github.com/parklab/ShatterSeek>, Akbani et al., 2015, Cell).

As suggested by Reviewer #1, we now refer to the observed catastrophic events as complex rearrangements dominated by segmental gains or dominated by losses, respectively, since no statistical framework is currently available to distinguish between chromoanasythesis and chromothripsis.

4, Statistical analyses were also missing in the current manuscript. Although the numbers of positive/total mice were provided in the Results section, it is hard to assess the significance of the observations.

As requested, statistical analyses were added on revised Figure 4 and on Supplementary Figures 2 and 6.

5, Many statements were only descriptive or even speculative. For example, "Cases with several normal copies and one larger derivative chromosome (presumably the copy affected by chromoanasythesis) suggest that polyploidization may precede chromoanasythesis, facilitating the survival of the cell after such a catastrophic event" provides no information but

may mislead readers because the authors neither clearly stated the phenomenon nor clarified all the behind possibilities.

In the revised version of the manuscript, the statements about the potential causative role of *Myc/Mycn* in catastrophic events were rephrased more carefully (formulated as a hypothesis, as suggested by Reviewer #1) and the related data were removed from the main figure. The observation about the putative order of events based on the M-FISH data was removed from the text.

Reviewers' Comments:

Reviewer #1:

Remarks to the Author:

The authors have done an outstanding job in responding to our comments and we recommend publication of the paper.

We have only two minor comments.

In the text the authors on page 6 state that p53^{-/-} Xrcc4 c⁻ mice have significantly less apoptosis. In supplemental figure 5, however, the graph indicates that there is no significant difference. Perhaps this is because of the intermediate time point, but this needs to be clarified before publication.

In Supplementary figure 3c, it seems that the y-axis labels (5 and -5) have been reversed.

I do not need to reevaluate the response to these minor points.

Reviewer #2:

Remarks to the Author:

My concerns and suggestions have been adequately addressed.

Reviewer #3:

Remarks to the Author:

- In the revised version of this manuscript, in an attempt to support their hypothesis on the contribution of DDR defects, the authors presented novel data on the characterization of other medulloblastoma mouse models. Interestingly /surprisingly, the models with loss of Rictor, in combination or not with loss of Pten, have a percentage of tumors with complex genomic rearrangements (50-100%) (Suppl fig.2) similar to what they observed in models with DDR defects (30-60%) (Fig1 c). These evidences somehow go against the author hypothesis of the contribution of the DDR components in the complex genomic rearrangements. While there are reports on the role of PTEN in DNA damage and repair, this reviewer is not aware of any role of Rictor in DNA damage response.

These new data, more than solving the issue of whether p53 loss alone is sufficient to induce complex rearrangements, add even more complexity, since model with supposedly DDR defects share a "lot in commons" with models the Brca2, Xrcc4 and Lig4 deletions.

How do the authors reconcile these data?

- From my original comments: "Supplementary Figure 10b, what are the levels of expression of the DNA repair genes in the samples with hemizygous deletion? Hemizigosity is not always associated with lower mRNA expression, there might be compensatory mechanism that increase their expression from the remaining allele.

Author response: As suggested, we now added the expression levels of the DNA repair genes in Supplementary Figure 10b as well as the significance analysis for differences in expression levels." Simply adding a plot as a supplementary figure without any comment, it's not that meaningful. My original question is referred to the fact that the association of hemizigosity of a specific gene with a specific phenotype should also be supported by expression data of that specific gene. The data presented Supplementary figure 10 b, show that there is no significant downregulation of the gene of interest and therefore it's really hard to interpret the claim that "In glioblastoma, loss of BRCA2, LIG4 or XRCC4 was associated with more frequent occurrence of complex genome rearrangements (Line 363)". The authors they should either test whether lower levels of BRCA2, LIG4 or XRCC4

are associated with increase in rearrangements or they should consider to remove the statement.

- In the discussion section on Myc and Mycn (line 444-458), the author should comment also on the fact that in the new mouse model that they have included, Myc overexpression IS NOT sufficient to induce catastrophic event (supplementary figure 2b). Different reviewers have specifically asked to the authors to be more conservative on this aspect.

Reviewer #4:

Remarks to the Author:

The authors have fully addressed my concerns.

Response to reviewers' comments

REVIEWERS' COMMENTS:

Reviewer #1 (Remarks to the Author):

The authors have done an outstanding job in responding to our comments and we recommend publication of the paper.

We have only two minor comments.

In the text the authors on page 6 state that p53^{-/-} Xrcc4 c^{-/-} mice have significantly less apoptosis. In supplemental figure 5, however, the graph indicates that there is no significant difference. Perhaps this is because of the intermediate time point, but this needs to be clarified before publication.

We modified the text (page 6) to mention at which time points the differences are significant and at which they are not.

In Supplementary figure 3c, it seems that the y-axis labels (5 and -5) have been reversed.

We corrected the y-axis labeling for Supplementary Figure 3c.

I do not need to reevaluate the response to these minor points.

Reviewer #2 (Remarks to the Author):

My concerns and suggestions have been adequately addressed.

Reviewer #3 (Remarks to the Author):

- In the revised version of this manuscript, in an attempt to support their hypothesis on the contribution of DDR defects, the authors presented novel data on the characterization of other medulloblastoma mouse models. Interestingly /surprisingly, the models with loss of Rictor, in combination or not with loss of Pten, have a percentage of tumors with complex genomic rearrangements (50-100%) (Suppl fig.2) similar to what they observed in models with DDR defects (30-60%) (Fig1 c). These evidences somehow go against the author hypothesis of the contribution of the DDR components in the complex genomic rearrangements. While there are reports on the role of PTEN in DNA damage and repair, this reviewer is not aware of any role of Rictor in DNA damage response. These new data, more than solving the issue of whether p53 loss alone is sufficient to induce complex rearrangements, add even more complexity, since models with supposedly DDR defects share a "lot in common" with models the Brca2, Xrcc4 and Lig4 deletions.

How do the authors reconcile these data?

The high prevalence of catastrophic events in the models based on *Rictor* inactivation in combination or not with *Pten* loss is of course interesting. We added one sentence to the discussion to mention this point (page 10).

We did not want to claim that DNA damage repair defects are the only source/cause of catastrophic events; Rictor and many other factors might very well play a role in complex genome rearrangements, which should be investigated in future studies.

- From my original comments: "Supplementary Figure 10b, what are the levels of expression of the DNA repair genes in the samples with hemizygous deletion? Hemizigosity is not always associated with lower mRNA expression, there might be compensatory mechanism that increase their expression from the remaining allele. Author response: As suggested, we now added the expression levels of the DNA repair genes in Supplementary Figure 10b as well as the significance analysis for differences in expression levels."

Simply adding a plot as a supplementary figure without any comment, it's not that meaningful. My original question is referred to the fact that the association of hemizigosity of a specific gene with a specific phenotype should also be supported by expression data of that specific gene. The data presented Supplementary figure 10 b, show that there is no significant downregulation of the gene of interest and therefore it's really hard to interpret the claim that "In glioblastoma, loss of BRCA2, LIG4 or XRCC4 was associated with more frequent occurrence of complex genome rearrangements (Line 363)". The authors they should either test whether lower levels of BRCA2, LIG4 or XRCC4 are associated with increase in rearrangements or they should consider to remove the statement.

As requested, we performed the significance tests regarding differences in RNA expression levels of *BRCA2*, *LIG4* and *XRCC4* depending on the presence of complex genome rearrangements. We added new plots in Supplementary Figure 10b showing these data (far right). We still think that the bar plot (left panel of Supplementary Figure 10b) is informative in terms of comparison of the murine models to human tumors; the lack of significance at RNA expression level may have many reasons and additional factors could play a role (e.g. expression of mutant versus wild-type allele, effect on protein expression...). However, we would of course remove panel 10b if the Editor considers that this is necessary.

- In the discussion section on Myc and Mycn (line 444-458), the author should comment also on the fact that in the new mouse model that they have included, Myc overexpression IS NOT sufficient to induce catastrophic event (supplementary figure 2b). Different reviewers have specifically asked to the authors to be more conservative on this aspect.

As suggested, we added this point to the discussion (page 11).

Reviewer #4 (Remarks to the Author):

The authors have fully addressed my concerns.